



# Relationship between Chemical Composition and Oxidative Potential of

# Secondary Organic Aerosol from Polycyclic Aromatic Hydrocarbons

Shunyao Wang[1], Jianhuai Ye[1], Ronald Soong[2], Bing Wu[2], Legeng Yu[1],

Andre Simpson[2], Arthur W.H. Chan[1,*]

[1]Department of Chemical Engineering & Applied Chemistry, University of Toronto

[2] Environmental NMR Centre, Department of Physical and Environmental Sciences,

University of Toronto Scarborough

*Correspondence to: Arthur W.H. Chan (arthurwh.chan@utoronto.ca)



**Abstract**
Owing to the complex nature and dynamic behaviors of secondary organic aerosol (SOA), its
ability to cause oxidative stress (known as oxidative potential, or OP) and adverse health
outcomes remain poorly understood. In this work, we probed into linkages between the chemical
composition of SOA and its OP, and investigated impacts from various SOA evolution pathways,
including atmospheric oligomerization, heterogeneous oxidation and mixing with metal. SOA
formed from photooxidation of the two most common polycyclic aromatic hydrocarbons
(naphthalene and phenanthrene) were studied as model systems. OP was evaluated using the
dithiothreitol (DTT) assay. The oligomer-rich fraction separated by liquid chromatography
contributed significantly to DTT activity in both SOA systems ($52\pm10\%$ for NSOA and $56\pm5\%$
for PSOA). Heterogeneous ozonolysis of NSOA was found to enhance its OP, which is
consistent with the trend observed in selected individual oxidation products. DTT activities from
redox-active organic compounds and metals were found to be not additive. When mixing with
highly redox-active metal (Cu), OP of the mixture decreased significantly for 1,2-
naphthoquinone ($42\pm7\%$), 2,3-dihydroxynaphthalene ($35\pm1\%$), NSOA ($50\pm6\%$) and PSOA
($43\pm4\%$). Evidence from proton nuclear magnetic resonance ($^{1}$H NMR) spectroscopy illustrates
that such OP reduction upon mixing can be ascribed to metal-organic binding interactions. Our
results highlight the role of aerosol chemical composition under atmospheric aging processes in
determining the OP of SOA, which is needed for a more accurate and explicit prediction of the
toxicological impacts from particulate matter.



## 1 Introduction

Exposure to particulate matter (PM) has been associated with various adverse health endpoints,

such as increased risks of myocardial infarction, ischemic heart disease, lung cancer,

exacerbation of asthma, and chronic obstructive pulmonary disease  (de Kok et al., 2006; Li et al.,

2003a; Li et al., 2003b; Nel, 2005; Risom et al., 2005; Thurston et al., 2016). As a result,

ambient $PM_{2.5}$ exposure ranks among the top 5 global mortality risk factors (Cohen et al., 2017).

Meanwhile, a decreased ambient PM level has been associated with longer life expectancies

(Pope et al., 2009). To establish causal links between aerosol exposure and health endpoints,

cytotoxic and carcinogenic potential has been investigated by epidemiological studies in the past

decades (Brunekreef and Holgate, 2002; Beelen et al., 2014; Lelieveld et al., 2015; Pope et al.,

2002), but the underlying mechanistic pathways by which PM causes adverse health outcomes

still remain poorly understood.

Oxidative stress has been proposed as one of the main mechanisms for PM toxicity in recent

years, and is often expressed as the oxidative potential (OP) (Li et al., 2003b). OP is the mass

normalized capacity of inhaled PM to induce oxidative stress, which is exhibited as redox

imbalance through consumption of antioxidants and generation of reactive oxygen species (ROS)

(Antiñolo et al., 2015; Shen et al., 2011; Shiraiwa et al., 2012). ROS include a variety of oxidants

such as superoxide ($O_2 \bullet^-$), hydroxyl radical ($\bullet OH$) and hydrogen peroxide (HOOH), which could

either be introduced into human body directly from inhaled PM or generated by targeted cells

(Nel et al., 1998; Pöschl and Shiraiwa, 2015; Rhee, 2006; Verma et al., 2015b). The generation

of ROS during multiphase interactions between air pollutants and human respiratory tract is

closely related to the chemical composition, since the combination of various pollutants may



influence chemical reactivity as well as bioavailability of PM while having synergistic or
nonlinear influences on its OP (Anti ñolo et al., 2015; Charrier et al., 2015; Fang et al., 2015;
Shiraiwa et al., 2012; Xiong et al., 2017).
OP of both organic and inorganic PM components have been evaluated by both cellular and
acellular assays. *In vitro* cellular assays were conducted by detecting biological endpoints of the
exposure, including heme oxygenase-1 (HO-1) and other cytokines as well as macrophage
related biomarker expressions (Krapf et al., 2017; Li et al., 2003b). On the other hand, acellular
assays use specific chemicals, such as dithiothreitol (DTT), ascorbate (AA) and glutathione
(GSH), as surrogates of low-molecular weight (MW) antioxidants (Fang et al., 2016; Godri et al.,
2011;McWhinney et al., 2013). Among acellular assays, the DTT assay quantifies OP by
measuring the DTT depletion rate over a fixed time interval, which mimics the physiological
process of electron transfer from biological antioxidants to dissolved $O_2$ (Cho et al., 2005). DTT
assay is one of the most commonly used OP evaluation methods, since DTT is a potent surrogate
for the total thio-pools (glutathione and protein thiols) while this assay can be conducted under
biologically relevant conditions (37 °C, pH=7.4) with relatively simple procedures (Cleland ,
1964; Hansen et al., 2009; McWhinney et al., 2011). OP levels measured by this assay have been
found to correlate well with cellular ROS expression as well as several airway inflammation
biomarkers, such as HO-1, tumor necrosis factor- α (TNF- α) and fractional exhaled nitric oxide
($FE_{NO}$) (Delfino et al., 2013; Li et al., 2003b; Tuet et al., 2017).

Secondary organic aerosol (SOA) from atmospheric oxidation of gaseous precursors comprises a
major fraction of submicron particulate matter. To date, The DTT assay has been applied in a
few studies to evaluate the OP of both laboratory and ambient SOA ( McWhinney et al., 2013;



Tuet et al., 2017b; Verma et al., 2015a). However, owing to the complex and dynamic property
of SOA, there is limited understanding of the relationship between detailed SOA composition
and its OP (Charrier and Anastasio, 2012;Pöschl and Shiraiwa, 2015; Tuet et al., 2017b). Tuet et
al. (2017b) studied the DTT activity of chamber-generated SOA from both biogenic and
anthropogenic VOCs under various conditions, showing that naphthalene SOA (NSOA) has the
highest OP.  Previous work (Antiñolo et al., 2015; Bolton et al., 2000; Charrier and Anastasio,
2012; Cho et al., 2005; Jiang et al., 2017; Tuet et al., 2017b) indicated while polycyclic aromatic
hydrocarbons (PAHs) are unreactive towards DTT while their oxidation products, such as
quinones, can be highly redox-active. Quinones can be directly emitted from traffic or formed
from secondary oxidation (Cho et al., 2004;McWhinney et al., 2013), and are able to consume
antioxidants in a catalytic cycle (Fig. 1) (Bolton et al., 2000;Valavanidis et al., 2005).
McWhinney et al.(2013) found three quinones (1,2- naphthoquinone, 1,4-naphthoquinone and 5-
hydroxy-1,4-naphthoquinone) in NSOA could only account 30 ±5 % for the observed OP of
NSOA, and the source of the remaining DTT activity remains unknown. Peroxides, which are
the major contributors to the OP of isoprene SOA (Jiang et al., 2017; Lin et al., 2016; Surratt et
al., 2010), may also be abundant in NSOA (Kautzman et al., 2009), but their contribution to the
OP of NSOA has not been evaluated.  In addition, the composition of SOA may evolve upon
atmospheric aging. Previous study (Verma et al., 2009) found ambient samples collected in the
afternoon had a larger fraction of water soluble organic carbon and higher OP, suggesting the
photochemical aging effects. There may also be correlations between the average carbon
oxidation state ($\overline{OS}c$) and OP of SOA at different stages of oxidation (Tuet et al., 2017a).
Moreover, mixing between organics and metals was also found to change the OP of specific
components in SOA (Xiong et al., 2017), but the mechanisms remain the tip of iceberg.




Here, we focused on understanding how the composition of PAH-derived SOA is related to its
strong OP (Tuet et al., 2017b). Specific questions we aim to address in this work are: what are
the compounds within SOA that are important for DTT activity? How does the OP change upon
atmospheric aging processes, including oligomerization, heterogeneous oxidation, and mixing
with transition metals (Gao et al., 2004; Rudich et al., 2007)? In this work, SOA from
photoxidation of naphthalene and phenanthrene were studied as model systems, and compared to
SOA from ozonolysis of α-pinene and limonene. The relative OP contributions of peroxides and
high-MW oligomers were evaluated. Effects of aerosol aging on OP was evaluated by examining
the OP of individual oxidation products known to be present in NSOA, and the OP of NSOA
samples that were further oxidized in the condensed phase. Lastly, the impacts of SOA mixing
with metal was explored by mixing SOA or redox-active SOA components with Cu (II), a
transition metal which has been identified with the highest OP in ambient particles (Charrier and
Anastasio, 2012). Additivity of OP revealed by DTT assay was investigated, and the mechanisms
of Cu-organic interactions were examined in detail using proton nuclear magnetic resonance
spectroscopy ($^1$H NMR).

**2   Methods**
**2.1 Flow tube experiments**
SOA was produced in a custom-built 10 L quartz flow tube. Details about the flow tube
conditions have been described in previous work (Ye et al., 2016; Ye et al., 2017). Prior to each
experiment, the flow tube was flushed with purified compressed air at a flow rate of 20 L min$^{-1}$
for over 24 hours.




To produce SOA from naphthalene or phenanthrene, solid PAH was placed in a heated container
(80 ℃), and the sublimed vapor was carried into the flow tube in a 0.2 L min$^{-1}$ flow of purified
compressed air. $O_3$ and water vapor were also added into the flow tube. $O_3$ was produced by
passing 0.5 L min$^{-1}$ oxygen (99.6%, Linde, Mississauga, Canada) through an UV $O_3$ generator
(No. 97006601, UVP, Cambridge, UK). Water vapor was produced by bubbling purified air
through a custom-made humidifier with a flow rate of 1.3 L min$^{-1}$. Residence time inside the
flow tube was maintained around 5 min. The flow tube was housed inside an aluminum
enclosure, which was equipped with 254 nm UV lamps (UVP, Cambridge, UK). The photolysis
of $O_3$ produces O ($^1$D), which react with water vapor to produce •OH and initiate photooxidation
of naphthalene/phenanthrene as well as SOA formation. During naphthalene/phenanthrene
photooxidation, $O_3$ concentration inside the flow tube was controlled around 1 ppm. In addition,
blank experiments were conducted under the same conditions, without injecting any
hydrocarbons.

Two types of SOA from monoterpene ozonolysis were also synthesized under similar conditions.
α-pinene (Sigma-Aldrich, 98%) or limonene (Sigma-Aldrich, 97%) was pre-dissolved in
cyclohexane (Sigma-Aldrich, 99.5%) with volumetric ratio of 1:500 or 1:1500, respectively. At
these ratios, the reaction rates between •OH and cyclohexane are at least a hundred time higher
than that of SOA precursors(Atkinson and Arey, 2003; Keywood et al., 2004). The experimental
solution was injected continuously into purified air flow by a syringe (1000 mL, Hamilton)
installed on a syringe pump (KDS Legato100) to achieve an initial concentration of 588±16 ppb
or 298±24 ppb of α-pinene or limonene, respectively. $O_3$ was produced by passing oxygen





through the $O_3$ generator at a flow rate of 0.2 L min$^{-1}$ or 0.3 L min$^{-1}$ for α-pinene or limonene,
respectively, such that the $O_3$ concentrations were at least 5 times higher than α-pinene or
limonene. Both experiments were conducted in the same flow tube without irradiation of UV
lights.

Temperature and relative humidity were monitored by an Omega HX94C RH/T transmitter. The
concentrations of SOA precursors at the inlet and outlet of the flow tube were measured by a gas
chromatography-flame ionization detector (GC-FID, Model 8610C, SRI Instruments Inc., LV,
USA) equipped with a Tenax® TA trap. Size distribution and volume concentration of SOA at
the flow tube reactor outlet were monitored using a custom-built scanning mobility particle sizer
(SMPS), which was composed of a differential mobility analyzer column (DMA, Model 3081,
TSI, Shoreview, MN, USA) with flow controls and a condensation particle counter (CPC, Model
3772, TSI, Shoreview, MN, USA). The SMPS data were inverted to particle size distributions
using custom code written in Igor Pro (Wavemetrics, Portland, OR, USA). By assuming a
particle density of 1.25 g cm$^{-3}$ for monoterpene SOA (Kostenidou et al., 2007; Shilling et al.,
2009) and 1.55 g cm$^{-3}$ for PAH SOA(Chan et al., 2009; McWhinney et al., 2011), volume
concentrations of particle were converted into mass concentrations and integrated as a function
of sample collection time and flow rates to obtain the total mass of collected SOA.

**2.2 SOA sampling and extraction**
All SOA samples were collected on 47 mm prebaked (500 ℃, 24h) quartz fiber filters (Pall, Ann
Arbor, MI, USA) in a stainless-steel filter holder after reaching a steady state yield, and then
wrapped in prebaked aluminum foil before being stored in sterile petri dishes sealed with





Parafilm M® at -20 ℃. Within 3 days of collection, the filters were extracted in methanol (HPLC
grade, 99.9%, Sigma Aldrich, St. Louis, MO, USA), by ultra-sonication at room temperature for
more than 3 minutes. After sonication, insoluble materials were filtered by a PTFE
(polytetrafluoroethylene) syringe filters (Fisherbrand™) with pore size of 0.22 µm. Chemical
analysis and DTT activities of the filter extracts were conducted within hours after extraction. As
negative control, filter samples were also collected during blank experiments and extracted in the
same manner.

**2.3 DTT assay**
OP of SOA and selected quinone/peroxide standards were quantified by the depletion rate of
DTT, an antioxidant that can be consumed by oxidative components in PM (Kumagai et al.,
2002). The protocols used in this work are adapted from those of McWhinney et al.(2011; 2013).
The SOA extracts were first evaporated to complete dryness in a 5.0 L min$^{-1}$ of N$_2$ using a blow-
off system (N-EVAP, Organomation, USA). Phosphate buffer (0.1 M, pH 7.4) was then added to
dissolve the SOA to achieve a concentration of 0.2 mM. For quinones, copper(II) sulfate and
peroxides, each pure compound was weighed and dissolved in 0.1 M phosphate buffer. The
concentration of quinones and copper (II) sulfate solutions was 1 µM, and the concentration of
each peroxide solution was 100 µM. The specific solution was then added into multiple wells in
a 96-well UV plate (Greiner Bio-One, Kremsmünster, AT), and immediately covered with
adhesive plate sealer (EdgeBio, Gaithersburg, USA). The plate was then placed in a UV-Vis
spectrophotometer (Spectramax 190, Molecular Devices Corporation, Sunnyvale, CA) for
incubation. The incubation temperature was maintained at 37 ℃ inside the spectrophotometer to
mimic human physiological conditions. 0.02 ml DTT was then added into each well to initiate



the redox-reactions. At each time point, 0.02 ml of 5, 5'-dithiobis (2-nitrobenzoic acid) (DTNB)
was added, which immediately consumed all the remaining DTT to form a yellow product, 2-
nitro-5-thiobenzoic acid (TNB) (Fig.S1). TNB was quantified by the light absorption at a
wavelength of 412nm, which was further converted to the DTT amount by calibration curve (Fig.
S2b) in order to obtain the DTT decay rate. The reaction was quenched by adding DTNB at
different wells at different times (all containing the same initial mixture), allowing for
quantification of the DTT decay rate over a 30 minute time interval (every 5 minutes for the first
10 minutes and then every 10 minutes). Blank control and the calibration curve for DTT
quantification are shown in Fig. S2. Here, we used the total DTT decay rate, $DTT_t$ ($\mu$M DTT
$min^{-1}$), to report the total oxidative capacity, as well as the mass normalized DTT decay rate,
$DTT_m$ (pmol DTT $min^{-1} \mu g^{-1}$ organics), to report the OP (Charrier et al., 2016; Jiang et al., 2017;
Xiong et al., 2017). Detailed information about the chemicals used in this assay is shown in
section S1.

**2.4 Quantification of peroxides in SOA**
Quantification of total peroxides in the four types of SOA was conducted using the iodometric-
spectrophotometric method, which quantifies total aerosol peroxides in all three forms ($H_2O_2$,
ROOH, ROOR) (Banerjee and Budke, 1964;Docherty et al., 2005):

202                   $$ROOR + 2I^- + 2H^+ \rightarrow 2ROH + I_2 \qquad (1)$$

203                   $$I_2 + I^- \rightarrow I_3^- \qquad (2)$$

where $I^-$ is oxidized to $I_2$ by peroxides under acidic conditions, and then complexes with
remaining $I^-$ to form $I_3^-$, a compound with brown color detected spectrophotometrically at a
wavelength of 470nm.  The concentration of each SOA solution was first adjusted to 5mM.  0.02





ml of potassium iodide solution (1 g ml$^{-1}$, KI dissolved in DI water), which provided the I$^-$ in
reaction (1), and 0.02 ml of formic acid (≥95%), which maintained the acidity, were added to
0.16 ml of the SOA solution in each well of a 96-well UV plate. The plate was immediately
sealed and incubated for 1 h following the same procedures as in the DTT assay, and the UV-vis
absorbance was measured at 470 nm. After testing for the sensitivities of various peroxides in KI
assay (Fig. S3), and following the previous work by Kautzman et al.(2009), benzoyl peroxide
(≥98%) was chosen to represent peroxides in NSOA and used as standards for mass calibration
in this study. All values are reported as mass fraction of peroxides in the total SOA.

**2.5 Heterogeneous oxidation**
Heterogeneous oxidation of NSOA was conducted by first cutting a filter with freshly collected
NSOA into halves (within 3 days of flow tube synthesis and stored at -20 ℃). One half of the
filter was placed in a sealed container, and an O$_3$ stream (~3ppm, from the previously mentioned
O$_3$ generator) was passed through the filter at a flow rate of 0.2 Lmin$^{-1}$. The other half of the
filter was treated in parallel with a 0.2 L min$^{-1}$ flow of N$_2$ over the same time intervals, to
account for evaporation and/or decomposition of SOA components at room temperature. 3 sets
of experiments were conducted with exposure times of 1 h, 12 h and 24 h where the O$_3$ exposure
can be determined by,
$$\text{O}_3 \text{ exposure} = \int_0^t [\text{O}_3]\,\mathrm{d}t = \langle \text{O}_3 \rangle_t \times t \qquad (3)$$

where $\langle \text{O}_3 \rangle_t$ is the time averaged O$_3$ concentration at a total flow rate of 0.2 L min$^{-1}$. The DTT
activity of each O$_3$-exposed aerosol sample was normalized to that of the corresponding N$_2$
exposure group.   Changes in organic carbon mass on NSOA filters exposed to O$_3$/N$_2$ for 1h, 12h



and 24h were also monitored with a thermal optical organic carbon/elemental carbon (OC/EC)
aerosol analyzer instrument (Sunset Laboratory Inc., Tigard, OR, USA). OC/EC content was
measured following the IMPROVE OC/EC protocol (Chow et al., 1993). Blanks were measured
before each run and subtracted from the sample measurements.

**2.6 Chromatographic Separation of NSOA**
To identify the relative contributions of monomers and oligomers in N/ PSOA to DTT activity,
the SOA extract was separated using ultra-high performance liquid chromatography (UHPLC),
and analyzed using electrospray ionization/Ion Mobility-Time of Flight Mass Spectrometry
(ESI/IMS-TOF MS, TOFWERK, Switzerland, hereafter referred to as IMS-TOF). SOA
methanol extract (30 g $L^{-1}$) was separated on a reverse phase column (ZORBAX Eclipse Plus
C18, Agilent, USA) with an initial mobile phase of 90% DI water and 10% HPLC methanol at a
flow rate of 0.15ml $min^{-1}$ (1290 Infinity II, Agilent, USA). The ratio of water to methanol was
gradually adjusted from 9:1 to 1:9 between 25 and 30 min. Separation temperature was set to 30 ℃
with a pressure setting of 150 bar. The outlet flow was regulated using a LC-MS post column
flow splitters (Supelco, SigmaAldrich, USA) at a ratio of 30:1. The major flow was collected in
two different fractions: the first fraction was collected between 6 and14 min, and the second
fraction was collected between 14 and 33 min for NSOA (3-17min and 17-28min for PSOA).
DTT assay was conducted on each fraction to assess their OPs. The minor flow injected into
IMS-TOF was controlled at 5ul $min^{-1}$ for mass spectrometric analysis in the negative mode.

A deactivated fused silica capillary (360μm OD, 50 μm ID, 50cm length, New Objective,
Woburn, MA, USA) was used as the sample transfer line between the UHPLC and the IMS-TOF.



The ESI source was equipped with an uncoated SilicaTip Emitter (360μm OD, 50 μm ID, 50 μm
tip ID, New Objective, Woburn, MA, US). Charged SOA droplets generated from the tip of the
emitter were transferred through a desolvation region by a 1L min$^{-1}$ $N_2$ flow at room temperature,
and ions produced from the evaporated droplets were introduced into the drift tube for ion
mobility separation. The IMS drift voltage was set to –1.2 kV for the negative mode. The
separation temperature was set to 80±1∘C with an operation pressure setting of 1.2 bar for the
whole mass spectrometer. After separation in the ion mobility region, the ion $m/z$ is measured by
high-resolution time-of-flight mass spectrometry within an $m/z$ range of 40 to 800. Resolution
(m/dm50) of the time-of-flight mass spectrometer is typically 3500–4000 FWHM at $m/z$ 250
(Groessl et al., 2015; Krechmer et al., 2016). Spectra recording and data processing of the current
study were performed using routines written in Igor Pro (6.37, Wavemetrics, OR, USA):
"Acquility" (version 2.1.0, http://www.tofwerk.com/acquility) for raw data acquisition and
"Tofware" (version 2.5.3, www.tofwerk.com/tofware) for post processing.

**2.7 $^1$H NMR spectroscopy**
$^1$H NMR spectroscopy was used to further investigate the mechanism behind the OP reduction
upon mixing specific organics and transition metals (Simpson et al., 2011;Simpson and Simpson,
2014; Smith and van Eck, 1999). NMR measurements were performed on a Bruker Avance III
NMR spectrometer (11.7 T), equipped with a 4 channel liquid state ($^1$H, $^{13}$C, $^{15}$N, $^2$H) inverse
detection probe (QXI) fitted with an actively shielded Z gradient. Typical parameters used for
1D $^1$H experiments were: a 9.5 μs $^1$H pulse, 64k acquisition points, 14 ppm spectral width and 8
transients were collected, with a total of 5.7s between scans. Before NMR analysis, each sample
was dissolved into deuterium oxide ($D_2O$, Cambridge Isotope Laboratories, 99.96 %) and





dimethyl sulfoxide (DMSO, Fisher Scientific, 99.9 %) at a ratio of 9:1. Here we used $D_2O$ as a
lock reagent by suppressing it through pre-saturation.

In order to monitor the duration for the nuclear spin magnetization returning to an equilibrium
state, NMR relaxation times T1(longitudinal direction), T2 (transverse direction) were analyzed.
T1 of the sample was measured through the standard inversion recovery experiment.  The delay
periods used for T1 measurements ramped from 0.001 s to 15s in 16 increments with a delay of
60s between scans, which represented > 5 x T1 time to permit full signal recovery. For each
delay period, 16 transients were collected. The T2 of the sample was measured through the
standard Carr-Purcell-Meiboom-Gill (CPMG) sequence. The delay periods used for T2
measurements ramped from 1.2ms to 614ms in 16 increments with a delay of 60s between scans,
which represented > 5 x T1 time to permit full signal recovery. For each delay period, 16
transients were collected. A total of 16 free induction decays  were collected for each of the
relaxation experiments, and relaxation time calculations were done on Bruker Dynamics Centre
(v 2.4.5) using mono-exponential fitting functions (Eq. 3 and 4 below).

291                             $f(t) = I_0 \times [1 - 2e^{-t/T1}]$                         (4)

292                             $f(t) = I_0 \times e^{-t/T2}$                         (5)

Eq. (4) and Eq. (5) are the fitting functions for T1 and T2, respectively, where "$I_0$" is the thermal
equilibrium state of the overall proton nuclear spin magnetization; "t" is the variable delay time.
All the $^1H$ NMR spectra were collected using TopSpin (Bruker, v 3.2). Post-NMR-data
processing was conducted in MestReNova (Mestrelab Resesarch, v 11.0.4) and Origin
(OriginLab, v 9).



**2.8 Statistical analysis**

Data in this study were interpreted as mean ±standard error of the mean (SEM, n=3), and

significance analyses among DTT activities were performed by Student's t-test with a 95%

confidence interval. A statistical value of $p < 0.05$ was considered significant.

**3 Results and discussion**

**3.1 DTT activity of laboratory generated SOA**

Here, we chose two types of SOA derived from PAHs, naphthalene and phenanthrene, as the

model SOA systems. The OP of NSOA has been shown to be the highest among various types of

SOA previously studied (Tuet et al., 2017a;Tuet et al., 2017b). At the same time, both NSOA

and PSOA contain quinones which are known to be highly redox active and exhibit high OP

(Cho et al., 2004; McWhinney et al., 2013). As a comparison, α-pinene and limonene SOA from

ozonolysis were chosen to represent biogenic SOA derived from monoterpenes. Experimental

conditions and SOA yield information are summarized in Table 1.

The mass-normalized DTT decay rate, DTTm, was applied here for OP evaluation (Fig. 2).

Similar DTTm have been reported for NSOA, with values of 118 pmol $min^{-1}\mu g^{-1}$ by McWhinney

et al. (2013) and 110 pmol $min^{-1}\mu g^{-1}$ by Tuet et al. (2017b) for NSOA generated from chamber

photooxidation under dry conditions. The DTTm of α-pinene SOA of 19.1±2.5 pmol $min^{-1}\mu g^{-1}$ in

this study is also consistent with those reported by Tuet et al.(2017b) and Jiang et al.(2017). The

similar values among the different studies highlight the reproducibility of results from the DTT

assay. To the best of our knowledge, the OP of PSOA and limonene SOA from this study are the

first ones reported in the literature. Both SOA derived from PAHs yield a higher OP than the two



types of monoterpene SOA. The similarities in OP between limonene and a-pinene SOA (cyclic
monoterpenes), and between naphthalene and phenanthrene SOA (PAHs) observed in this study
further confirms the hypothesis proposed by Tuet et al.(2017b) that the intrinsic OP of SOA is
closely related to the molecular skeleton of the precursor.

One of the reasons for the high OP exhibited by PAH-derived SOA is the abundance of redox-
active quinone moieties in SOA compounds (Lee and Lane, 2009). The cytotoxic and
carcinogenic effects from quinone-like compounds are well recognized in the field of
biochemistry (Bolton et al., 2000; Valavanidis et al., 2005), and the toxicity of PM has been
attributed to the presence of quinones. Charrier and Anastasio (2012) have found the OP of
several quinones are comparable to transition metals in ambient particles. The importance of
quinone-like components to OP was also evaluated by examining the changes in DTT activity in
response to the presence of 2,4-dimethylimidazole, which has been shown to be the co-catalyst
of the quinone redox cycle (Jiang et al., 2017). However, McWhinney et al. (2013) quantified
three quinones in NSOA using GC/MS and found that these quinones can only account for 30%
of the total NSOA DTT response. The remaining DTT activity may arise from other quinone-like
compounds that have not been identified, or from other oxidation products in NSOA. Given this
knowledge gap, we examine the potential roles of peroxides, oligomers and other more
oxygenated products that may explain the high DTT activities of NSOA in the next sections.

**3.2 OP contribution from peroxides**
One of the main hypothesis in this study is that organic peroxides contribute to OP. Organic
peroxides have been identified to be major components in both laboratory and ambient OA



(Jokinen et al., 2014; Lin et al., 2016; Surratt et al., 2010; Zhang et al., 2015; Zhang et al., 2017).
They can play important roles in forming high-MW oligomers (Docherty et al., 2005) and highly
oxygenated molecules (Mentel et al., 2015). Recent studies have shown that peroxides may also
be important for OP. Kramer et al. (2016) suggested that isoprene-derived
hydroxyhydroperoxide (ISOPOOH) is an essential contributor to the OP of isoprene SOA,
consistent with the results of bulk peroxide measurements using 4-nitrophenylboronic acid assay
(NPBA assay) by Jiang et al. (2017). Since peroxides have been proposed to be a major
component in NSOA (Kautzman et al., 2010), it is essential to determine whether or not these
peroxides can account for the remaining OP contribution (McWhinney et al., 2013).

Here we compare the NSOA and α-pinene SOA systems to determine the role of peroxides in OP.
The KI assay is known to be sensitive to all types of SOA peroxides (ROOR, ROOH and HOOH)
(Banerjee and Budke, 1964), and we confirmed its sensitivity by conducting calibrations with 4
different peroxides (Fig. S3). Similar KI response factors were observed with hydrogen peroxide,
cumene hydroperoxide, tert-butyl peroxide and benzoyl peroxide. Since it is likely that the
peroxides in NSOA have one aromatic ring are mostly in the form of ROOR (Kautzman et al.,
2009), we used benzoyl peroxide as the mass calibration standard. The mass fraction of
peroxides and DTTm of each SOA system are shown in Table 2. A high percentage of peroxide
(40-100%) was observed in α-pinene SOA, which was consistent with the results (47%) in the
study by Docherty et al. (2005). Meanwhile, a very low percentage of peroxides (<3%) was
found in NSOA system, a result that is inconsistent with previous work (>20%) by Kautzman et
al. (2009). The difference in measured peroxide content is most likely due to the difference in the
UV light source. Kautzman et al. (2009) and McWhinney et al. (2013) used $H_2O_2$ photolysis



under black lights (~350 nm), whereas in our study 254 nm UV lamps were used to photolyze $O_3$
and generate $O\ (^1D)$. Organic peroxides in SOA are known to be photo-labile (Banerjee and
Budke, 1964; Krapf et al., 2016; Wang et al., 2011) and had likely decomposed rapidly under the
shorter UV wavelengths used in our studies. Despite the differences in light conditions and
peroxide content, the DTTm measured for NSOA in this study is consistent with those measured
in two separate studies (McWhinney et al., 2013; Tuet et al., 2017). Also, the $DTT_m$ of NSOA
was found to be significantly higher than that of α-pinene SOA, which contains a large fraction
of peroxides. Therefore, from our work, there is no evidence showing that peroxides contribute
significantly to the high $DTT_m$ observed in NSOA. Even if organic peroxides were present at a
mass fraction of around 20%, as reported by Kautzman et al., (2010), we expect these peroxides
would react with DTT at a similar rate as benzoyl peroxide, which has a similar structure that the
proposed peroxides. The DTTm of benzoyl peroxide (ROOR-type) is 38 pmol $min^{-1}ug^{-1}$, which
is around 3 times lower than the DTTm of NSOA. It should also be noted that organic
hydroperoxides are the major OP contributors for biogenic SOA, such as isoprene and
monoterpene SOA (Jiang et al., 2017). One of the potential mechanisms is the formation of
hydroxyl radicals from the decomposition of organic hydroperoxides in water (Tong et al., 2016).
For the NSOA system, our results suggest that other non-peroxide species are likely to serve as
major contributors to OP.

**3.3 OP of oligomers in NSOA**
Atmospheric OA also undergoes extensive oligomerization, forming high-MW compounds that
have profound impacts on SOA physicochemical properties (Hallquist et al., 2009; Rudich et al.,
2007; Trump and Donahue, 2014;Wang et al., 2011). Laboratory photooxidation of aromatic



compounds produces a substantial fraction of oligomers in the SOA (Kalberer et al., 2004) and
these oligomers may be highly functionalized (Gao et al., 2004; Tolocka et al., 2004;). IMS-TOF
analysis reveals that a substantial fraction of the signals in NSOA and PSOA are located in the
high *m/z* range, which are associated with high-MW oligomeric products. Since previous studies
have largely focused on monomeric quinones (such as 1,2-naphthoquinone or 9,10-
phenanthrenequinone), the contribution of high-MW products to OP have not been studied and
may explain the "missing" OP contributors.

To evaluate OP of high-MW products in NSOA and PSOA, solutions of SOA extract were
separated in a C18 reverse phase column into two major fractions. As shown in Fig. 4, when
analyzed by IMS-TOF, the first fraction was found to contain relatively higher signals at *m/z*
associated with monomers, and the second fraction contain products with higher signals located
in a higher *m/z* range. It should be noted that complete separation could not be achieved in this
work. Other techniques, such as size exclusion chromatography (Di Lorenzo and Young,
2016;Di Lorenzo et al., 2017), may yield better separation based on molecular weights, but may
not be able to resolve compounds in the relative low molecular weight range in the current study.
Nonetheless, the first fraction can be qualitatively described as a "monomer-rich" fraction, and
the second fraction can be regarded as an "oligomer-rich" fraction (Fig. 4, Fig. 5a, b). The DTT
assay was then conducted on both of the original SOA solution as well as the two separated
fractions. Since the amount of organic material in each fraction is not known, we use the total
DTT activity (DTTt, in μM min$^{-1}$) to qualitatively compare the oxidative capacities of the two
fractions.



As shown in Fig. 5, both the monomer-rich and oligomer-rich fractions are reactive towards DTT.
For NSOA, the OP contribution from the monomer-rich fraction and the oligomer-rich fraction
were 16±3% and 56±10%, respectively (Fig. 5c). For PSOA, the OP contribution from the
monomer-rich fraction and the oligomer-rich fraction were 40±8% and 50±5%, respectively (Fig.
5d). In both systems, the oligomer-rich fraction caused a more rapid decay in DTT than the
monomer-rich fraction even with a lower summed ion signal of low-MW constituents (Fig. 5a, b
for NSOA and PSOA, respectively). These qualitative results suggest that while the current focus
of health studies has been focused on monomeric quinones, other higher-MW products may be
important for the OP of NSOA and PSOA. Specific molecular characteristics of these high-MW
OP contributors are currently unknown, and understanding them will be the subject of future
research. It is very likely that the oligomers also contain redox-active quinone functional groups,
such as those formed on the surface of oxidized soot (Antiñolo et al., 2015), and are therefore
important for OP.

**3.4 OP from heterogeneous oxidation**
In addition to its complexity, the composition of SOA is also highly dynamic and evolves upon
atmospheric oxidation (Jimenez et al., 2009). Heterogeneous oxidation in the particle phase is
one of the major pathways in aerosol aging (George and Abbatt, 2010;Rudich et al., 2007) and
can increase oxygen content during the functionalization processes. McWhinney et al. (2013)
attributed 21% of the NSOA's DTT activity to two quinone isomers (1,2-NQN and 1,4-NQN) in
NSOA while found a higher DTT contribution (30%) when they took 5-hydroxy-1,4-
naphthoquinone (5-OH-1,4-NQN) into consideration. Thus, we expect that oxygenated



derivatives produced upon heterogeneous oxidation may also contribute to the OP of SOA, and
the OP of SOA could be enhanced by heterogeneous oxidation.

We first examined the changes in OP with additional functional groups in known organic
compounds. As shown in Fig. 6, two pairs of organic compounds in NSOA were chosen: 1,4-
NQN, and 5-OH-1,4-NQN were used to study quinone-like compounds while naphthol (NPL)
and 1,3-dihydroxy naphthalene (1,3-DHN) were used to compare phenol-like compounds. The
$\overline{OS}$c of those four components were calculated (Kroll et al., 2011), as shown in Fig.6. Our results
demonstrated a higher DTTm for standards with higher oxidation states. For each addition of an
OH group to the selected molecule, the OP increases. OP of an aromatic compound is therefore
shown here to be associated with its degree of oxygenation and is demonstrated here
fundamentally using individual organic compounds.

More broadly, oxidation also increases the degree of oxygenation in the bulk aerosol phase, and
increases OP. Here we conducted heterogeneous oxidation by exposing filter-collected SOA to
$O_3/ N_2$ with the same flow rate. The $N_2$ exposure group is used as the control group in order to
isolate the effects of evaporation and/or decomposition at room temperature from those of
heterogeneous $O_3$ oxidation. For each of the exposure (1h, 12h, 24h), the DTTt of $O_3$ exposure
group was normalized by the DTTt of the corresponding $N_2$ exposure group (Fig.7a). OC loss
was determined by thermal optical OC/EC analysis, and was observed to be 17% and 13% for
the $O_3$ and $N_2$ exposure groups after 24 hours, respectively (Fig.7b). Generally, DTTt of NSOA
filter under $O_3$ exposure was higher than that of $N_2$ exposure. The study of Anti ñolo et al.(2015)
also showed an increased redox activity of soot accompanied by an increased amount of



oxygenated derivatives (quinone) under heterogeneous oxidation. However, the enhanced
oxidative capacity from heterogeneous ozonolysis appeared to decrease with longer exposure to
$O_3$ (Fig.7a), which we hypothesize may result from functionalization as well as fragmentation of
organic molecules during heterogeneous oxidation (Kroll et al., 2009). This can be further
confirmed with the observed changes in the different OC fractions, as shown in Fig. S4. Within
the 24h exposure, the volatile fractions (OC1 and OC2) of the $O_3$ exposed group increased while
the less volatile fractions (OC3 and OC4) decreased compared to the $N_2$ group, which suggested
the decomposition of high-MW (low volatility) species into low-MW (high volatility)
compounds. Previous work has also shown fragmentation can play a dominant role in a late stage
of heterogeneous oxidation (Kroll et al., 2011). The overall increased volatility may lead to
evaporation of smaller redox-active molecules and decrease the DTTt compared to the $N_2$
exposure group. It should also be noted that the $O_3$ concentrations to which NSOA are exposed
here are about 100 times higher than typical ambient levels (Finlayson-Pitts and Pitts Jr, 1999).
Assuming heterogeneous oxidation mechanisms are linear and an ambient $O_3$ concentration of 30
ppb, $O_3$ exposure for 1h, 12h, and 24h under our experimental conditions represent 4, 50, 100
days of aging in the atmosphere. Therefore, we anticipate an overall enhancement in OP under
ambient conditions. Though similar observations have been made in soot (Antiñolo et al., 2015)
and diesel exhausted particles (McWhinney et al., 2013), this enhancement in OP by
heterogeneous oxidation is shown here for the first time in SOA particles.

**3.5 OP changes upon mixing with Cu**
Ambient PM forming from mixed sources is frequently composed of both organics and metals.
To date, both organics (quinone) and transition metals (Cu, Fe, Mn etc.) have been shown to be



redox-active (Charrier and Anastasio, 2012; Xiong et al., 2017). Metals in ambient particles can
range from insoluble substances to soluble cations, leading to various health outcomes after
deposition onto the human respiratory tract (Gojova et al., 2007; Oberdörster et al., 2005).

Based on the chemical composition and the assumption that DTT activities of quinones and
transition metals are additive, previous studies have attempted to reconstruct the overall OP in
ambient particles based on the chemical composition (Charrier and Anastasio, 2012;Charrier et
al., 2015). However, addition of a transition-metal chelator did not result in significant changes
in the expression of inflammatory biomarkers (Donaldson et al., 2001), suggesting that oxidative
activities from different transition metals may not be additive. In our study, significant
reductions in OP were observed for PSOA ($43\pm4\%$) and NSOA ($50\pm6\%$), when they were mixed
with Cu (II) (Fig.8a). Conversely, no significant OP reduction was observed when α-pinene or
limonene SOA was mixed with Cu (II). To further investigate the cause of this reduction, we
examined the OP of 1,2-NQN, 1,4-NQN, 1,3-DHN and 2,3-DHN and the effects from mixing
with Cu(II). Significant OP reductions for 1,2-NQN ($42\pm7\%$) and 2,3-DHN ($35\pm1\%$)(Fig. 8b)
were observed, but no such changes were observed with 1,4-NQN or 1,3-DHN upon metal
mixing. It should be noted that a mixture of phenanthrenequinone and Cu(II) did not show a
significant reduction in DTT activity in a study by Charrier and Anastasio (2012) while it is
likely that  the DTT measurements may be affected by the inefficiency of the quench regent,
trichloroacetic acid (Curbo et al., 2013). Furthermore, an increased level of OP reduction was
observed when an increasing amount of Cu (II) were mixed with the same amount of 1,2-NQN
(Fig. S6). Based on the observation that OP reduction occurs only when there are neighboring
oxygenated functional groups, we hypothesize that the OP reduction is related to formation of



covalent bonds between the electron-deficient Cu (II)  and the electron-donating polar functional
groups. The formation of quinone-copper complexes have been demonstrated previously
(Dooley et al., 1990; Klinman, 1996), and may be responsible for reducing the overall OP.

To understand the underlying mechanism, [1]H NMR spectroscopy was used to monitor the
formation of the organic-Cu complex. [1]H NMR has previously been applied to study the binding
between metals and organics (Peana et al., 2015; Syme and Viles, 2006). The relaxometry (T2)
of NSOA illustrated in Fig. 9 shows a decreasing trend on average for T2 relaxation time when
Cu (II) was added to the system. Such decrease in relaxation time indicates interactions between
copper and SOA components.  Due to the complexity in the SOA NMR spectra, it is still
currently challenging to specifically identify binding between individual NSOA components and
Cu. To further investigate the binding reactions, [1]H NMR measurements were made for the
compounds present in NSOA, as previously mentioned, in the presence and absence of Cu, as
shown in Fig. 10. Significant interactions between Cu and 1,2-NQN are evidenced by the
broadened peak shapes (Fig. 10a),  caused by the coordination of adjacent oxygen groups with
the copper ion, which has also been well documented in previous studies with similar
compounds (Inoue and Gokel, 1990;Schmidt et al., 1990;Tolman, 1977). Protons proximal to the
binding sites are more significantly broadened, while protons further away are less affected.
Conversely, mixing of 1, 4-NQN with Cu led to little changes in peak shape, indicating the lack
of any interactions with copper, as shown in Fig. 10b. Similar phenomena were also observed
with another pair of isomers (2, 3-DHN and 1,3-DHN). As shown in Fig.10c, 2, 3-DHN shows a
clear change in peak shape indicating that these hydroxyl moieties on adjacent carbons are very





important for copper coordination. On the other hand, the 1, 3-DHN structure shows very little
peak broadening when mixed with copper.

Such binding evidence was further supported by NMR relaxation data (Fig. 11). As copper is
paramagnetic, it is an effective relaxation agent, and protons brought into its proximity undergo
faster T1 and T2 relaxation that manifest themselves as spectral broadening in 1D NMR (Fig. 10).
For the epitope maps, the largest circles indicate the least interactions with copper. For both T1
and T2 data (Fig. 11, Table S1), protons adjacent to the binding sites a and f (pronto peak
assignment based on Fig.10a, Fig.S5) underwent significant changes in relaxation indicating the
metal coordinating to the neighboring oxygen groups. Proton e changed less as it is located
further away from the copper binding site. While copper has a relatively mild effect on protons b
and d, a significant reduction in T1 and T2 for proton c was observed. Such interesting
phenomenon is due to the increased nuclear Overhauser effect (NOE) from the b and d sites. In
the absence of binding, protons b and d would in part relax via NOE with protons a and e, which
relax rapidly due to the copper binding. As such protons b and d can no longer lose
magnetization via an Overhauser effect owing to a and e. Instead, they pass magnetization to
position c which underwent enhanced level of relaxation as a result. The NOE effect for ring
systems with similar structure has also been demonstrated by several previous publications
(Kowalewski and Maler, 2006; Rehmann and Barton, 1990). It should be noted that the increased
level of OP depletion is accompanied by an increased ratio of Cu to 1,2-NQN (Fig. S4a), the
degree of broadening in the 1D NMR proton peaks becomes more significant (Fig. S4b), and the
relaxation times T1, T2 decreases (Fig. 11). All these observations together illustrate that the
reduction of OP is proportional to the binding between Cu and organics, and further supports the



mechanism behind OP depletion in the DTT assay. Based on the conclusions from the individual
organic standards, the overall decrease in relaxation times for NSOA mixing with Cu (shown in
Fig. 9) likely indicates that Cu(II) are binding with NSOA components, limiting the redox
activities and OP of both the Cu(II) ions and the redox-active NSOA components.

**4  Implications**
Oxidative stress caused by ROS production and antioxidant consumption is one of the most
commonly studied mechanisms for PM toxicity (Nel, 2005;Rhee, 2006;Manke et al., 2013). Here
we performed OP evaluation of two SOA formed from PAHs (naphthalene, phenanthrene) by the
DTT assay, and investigated the linkages between SOA OP and chemical composition upon
various atmospheric aging processes. SOA derived from ozonolysis of monoterpenes (α-pinene,
limonene) have a lower DTTm than that of the two PAHs derived SOA, which could be
attributed to the high redox-active quinone-like components. This is also consistent with the
previous hypothesis that OP of SOA is highly dependent on the identity of its precursor (Tuet et
al., 2017a). To further link SOA OP to its chemical composition, this study also explored the
possible impacts from atmospheric aging processes so as to provide mechanistic understanding
for ambient observations.

Over the span of atmospheric lifetime, the mass and chemical composition of SOA can be
affected by aging processes (Kroll et al., 2009; Lim et al., 2017). The aerosol aging processes
that we studied here include oligomerization, heterogeneous oxidation and metal mixing. Apart
from quinones that are well known to exhibit high OP in aerosol samples, OP contributions from
peroxides in our NSOA system are likely to be insignificant. Rather, oxygenated derivatives



were shown here to contribute greater OP than their precursors in our study of selected organic
individuals, and heterogeneous oxidation of NSOA was shown to lead to greater OP as well.
Moreover, DTT activities of the monomer-rich and oligomer-rich fractions in NSOA separated
by liquid chromatography showed oligomers are OP contributors in SOA. While organic
peroxides have been proved to be very labile components with half-lives of minutes at room
temperature (Krapf et al., 2016), SOA oligomers are relatively stable and highly oxygenated with
their ratio of total organic molecular weight per organic carbon weight (OM:OC) similar to that
of atmospheric humic-like substances (HULIS)(Altieri et al., 2008). Consistent with OP
contributions from oxygenated components enriched HULIS fraction in ambient PM (Verma et
al., 2015b), this study also shows evidence for OP contributors from atmospheric aging products
of PAH-derived SOA, indicating major organic PM OP contributors could be less volatile than
previously thought, and may more readily remain in the particle phase under atmospheric aging.
Nevertheless, future work should focus on improving separation methods, allowing for more
precise measurements of OP from SOA oligomeric constituents.

A reduction in OP was observed when mixing NSOA/PSOA with Cu (II), resulting in a non-
additive effect. However, no such reduction was observed in α-pinene SOA or limonene SOA.
Using $^1$H NMR spectroscopy, we demonstrate that the reduction in OP is likely caused by
binding between Cu (II) and redox active organic compounds. Both the peak broadening in 1D
NMR spectra and shorter relaxation times are observed for compounds that exhibited OP
reduction (1,2-NQN, 2,3-DHN) upon Cu(II) mixing. Additionally, the greater amount of Cu (II)
mixed in, the enhanced OP reduction and the decrease in relaxation times showed up. While it is
still challenging to determine which NSOA components are binding with Cu (II), the overall





relaxation time also decreased when NSOA was mixed with Cu(II), indicating binding between
Cu(II) and various NSOA components. Based on our work, previously recognized redox-active
organic and inorganic components in ambient particles (Charrier and Anastasio, 2012;
McWhinney et al., 2011; Monks et al., 1992;  Turski and Thiele, 2009) may bind with each other
once mixed during atmospheric aging processes (external) or within the physiological
environment of the human body (internal). The current study demonstrates that such binding
leads to a lower OP, which may be relevant to many health outcomes. In the future, a more
detailed understanding of SOA binding with metal components and the effects on the oxidative
health outcomes will be essential. It should also be noted that the DTT assay alone may not be
entirely representative of physiological ROS variations (Tuet et al., 2017a; Xiong et al., 2017;).
More *in vitro* and *in vivo* work should be performed in establishing the relationship between
chemical composition and the OP of aerosol.

ASSOCIATED CONTENT
**Supporting Information**.

AUTHOR INFORMATION
**Corresponding Author**
* Address: 200 College Street, Toronto, ON, M5S 3E5

615        Email: arthurwh.chan@utoronto.ca

Phone: +1 (416)-978-2602
**Notes**
The authors declare no competing financial interest.




**ACKNOWLEDGEMENT**

This work was supported by Natural Sciences and Engineering Research Council Discovery Grant, Canadian Foundation for Innovation John R Evans Leaders Fund, and the Ontario Early Researcher Award. The authors would like to thank Dr. Jon Abbatt, Dr. Greg Evans and Manpreet Takhar for helpful discussion.

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



**Table 1. Flow tube experimental conditions**

| Compund | Reaction[a] | ΔHC ppb | Y[b,c] % | RH |
|---|---|---|---|---|
| limonene | ozonolysis | 251±23 | 25±3.9 | 14±1% |
| α-pinene | ozonolysis | 304±17 | 19±4.2 | 14±1% |
| naphthalene | photooxidation | 6436±402 | 28±6.7 | 57 ±5% |
| phenanthrene | photooxidation | 4050±578 | 12±2.6 | 57 ±5% |

a. Temperature in all experiments is around room temperature (22-25°C).
b. SOA mass yields were calculated without particle wall loss correction.
c. SOA density in this study was assumed to be 1.25 g cm$^{-3}$ for monoterpene SOA (Shilling et al., 2009), and 1.55 g cm$^{-3}$ for PAHs SOA(Chan et al., 2009).

**Table 2. SOA peroxide content and OP**

| Organics | Peroxide percentage % | DTTm pmol min$^{-1}$ µg$^{-1}$ |
|---|---|---|
| Naphthalene SOA | <3% | 100-129 |
| α-pinene SOA | 40-100% | 10-20 |
| Benzoyl peroxide | 100% | 160 |



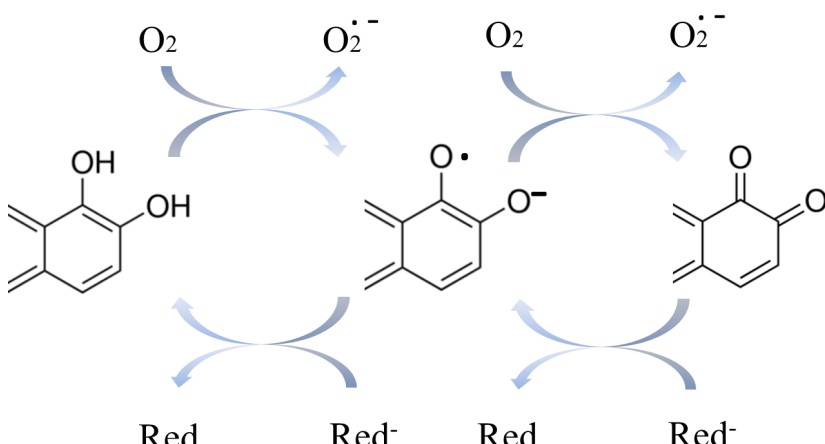

**Figure 1.** A simplified mechanism of redox cycling of quinone-like substances and formation of superoxide anion radicals. **_Red_** refers to a general reductant. In the redox cycle, regenerated quinone serves as a chemical intermediate to transfer electrons from reductants to oxygen to form superoxide ($O_2^{\cdot -}$).



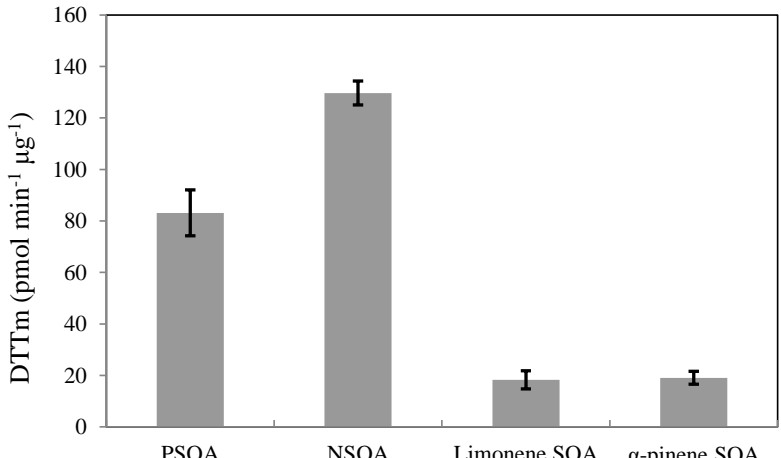

**Figure 2.** DTTm (pmol min⁻¹ µg⁻¹) for SOA formed from various types of hydrocarbons

(phenanthrene, naphthalene, limonene and α-pinene). Each measurement was conducted in

triplicates, and the error bar represents the standard error of the mean (SEM).





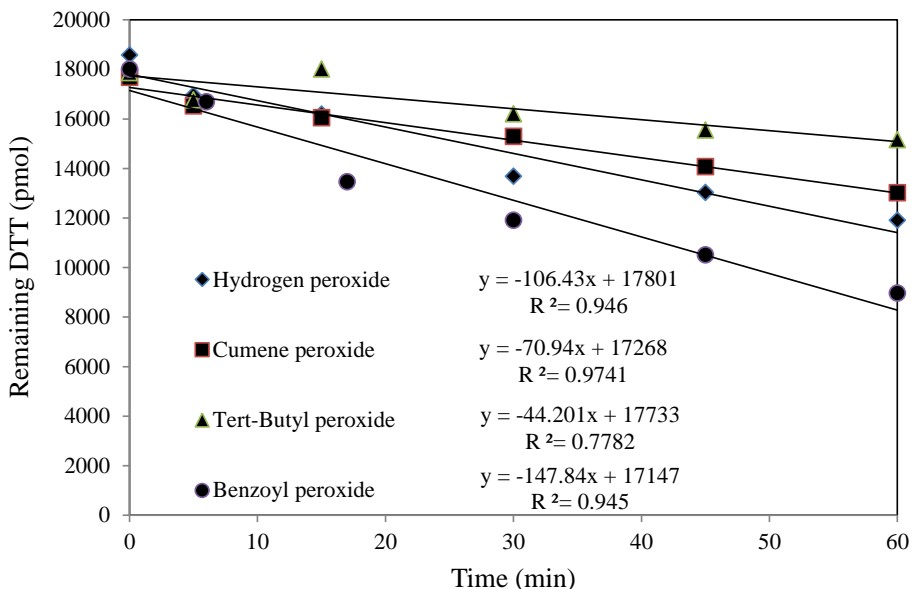

**Figure 3.** DTT activity of various types of peroxides (hydrogen peroxide, cumene peroxide, tert-Butyl peroxide, benzoyl peroxide). With the same initial concentration of peroxide (0.1mM), benzoyl peroxide has the highest DTT activity (147.8 pmol min$^{-1}$, which can be converted to DTTm of 38 pmol min$^{-1}$ ug$^{-1}$).




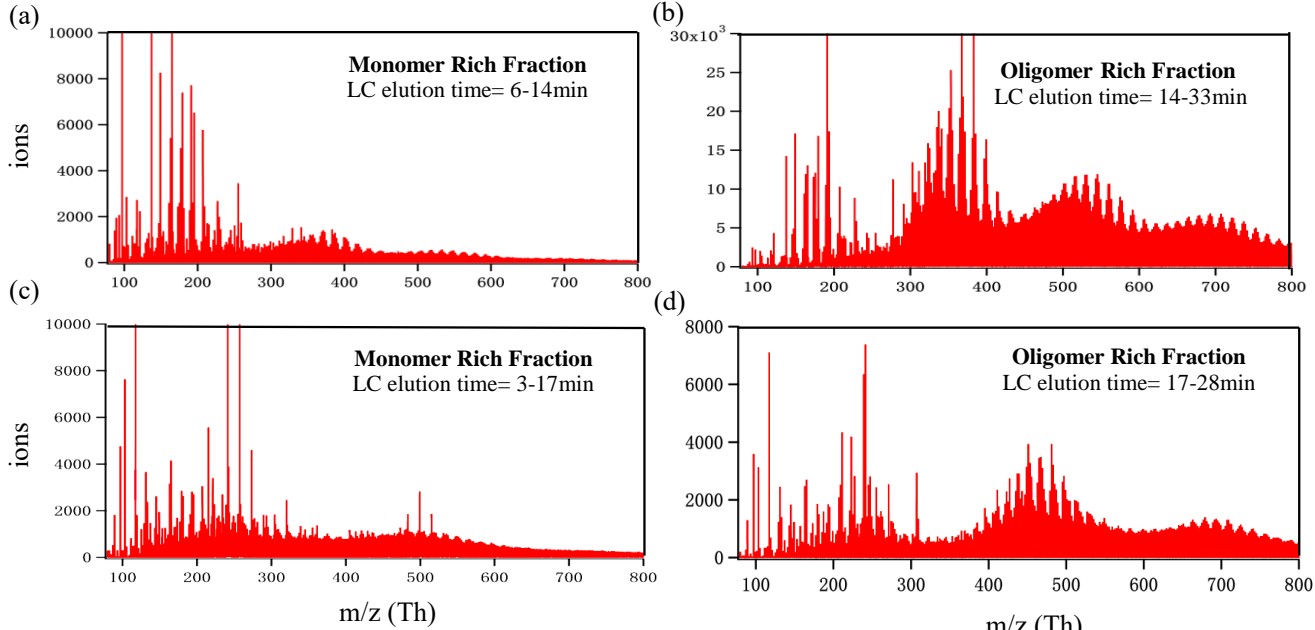

**Figure 4.** IMS-TOF mass spectra for (a) monomer-rich fraction (b) oligomer-rich fraction in

NSOA and (c) monomer-rich fraction (d) oligomer-rich fraction in PSOA. During a total of 46-

min elution, the majority of NSOA monomers eluted at 6-14 min, and most of the oligomers

eluted at 14-33min. For PSOA system, the majority of monomers eluted at 3-17 min, and most of

the oligomer- rich fraction eluted at 17-28 min.



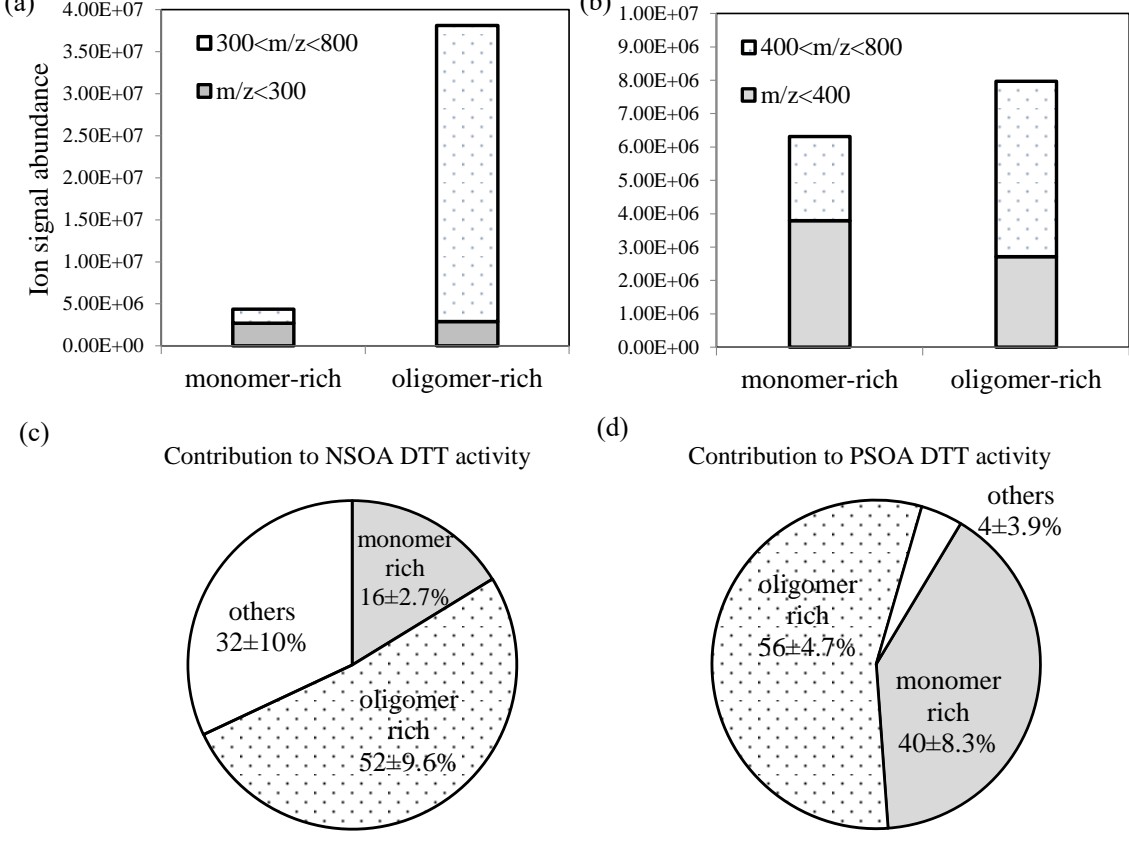

**Figure 5.** Sum of ion signals for monomers and oligomers in monomer-rich fraction and

oligomer-rich fraction for NSOA (a) and PSOA (b) systems. OP (DTTt) contributions from

monomer-rich fraction and oligomer-rich fraction in NSOA (c) and PSOA (d) systems. The

remaining DTT activity (others, white) is attributed to residual SOA fractions that did not clearly

elute out.



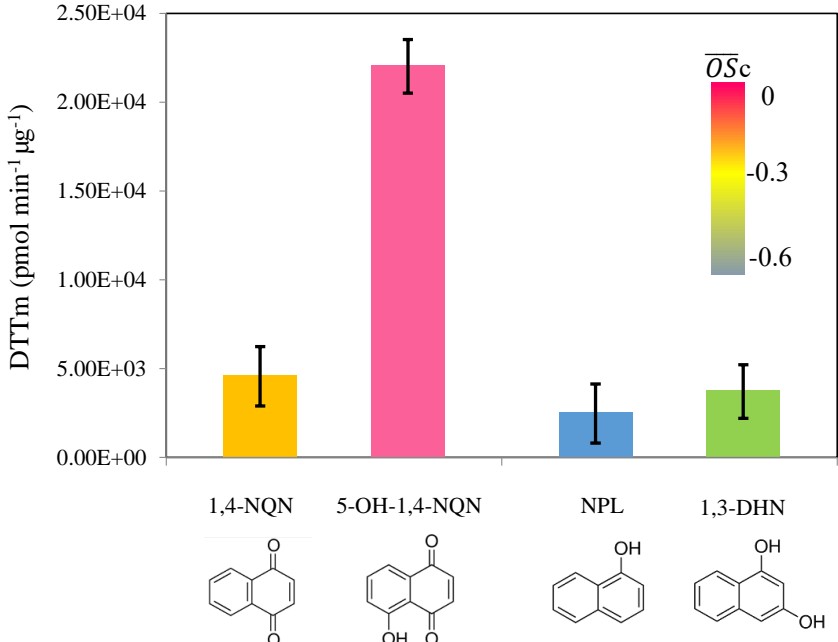

**Figure 6.** DTTm for two selected pairs of oxygenated derivatives in NSOA system (1,4-NQN vs.

5-OH 1,4-NQN, NPL vs. 1,3-DHN). Averaged carbon oxidation state ($\overline{OS}$c) of each component

is shown in color (color scale shown on top-right). Each measurement was conducted in

triplicates, and the error bar here represents the SEM. The asterisk indicates significant

difference between each pair of measurements at the 95% confidence level.



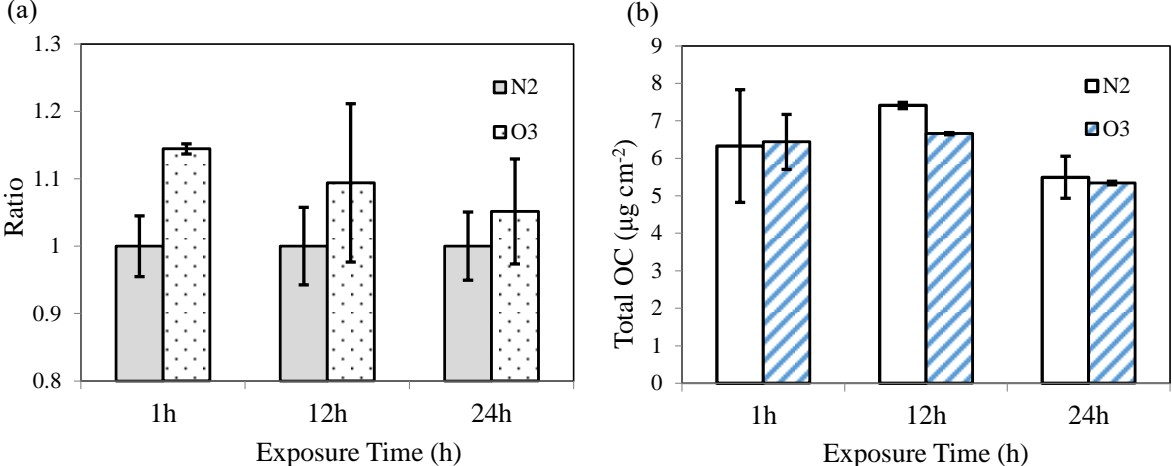

**Figure 7.** (a) Relative DTTt for NSOA after $O_3$ and $N_2$ exposure. For each of the exposure duration (1h, 12h, 24h), the DTTt of $O_3$ exposure group was normalized by the DTTt of the corresponding $N_2$ exposure group. Generally, DTTt of NSOA that underwent heterogeneous ozonolysis was higher than that of the $N_2$ control. (b) OC/EC measurement results show total OC mass loss (17% and 13% for $O_3$ and $N_2$ exposure, respectively) after 24-hour exposures. Each measurement was conducted in triplicates, and the error bar represents the SEM.



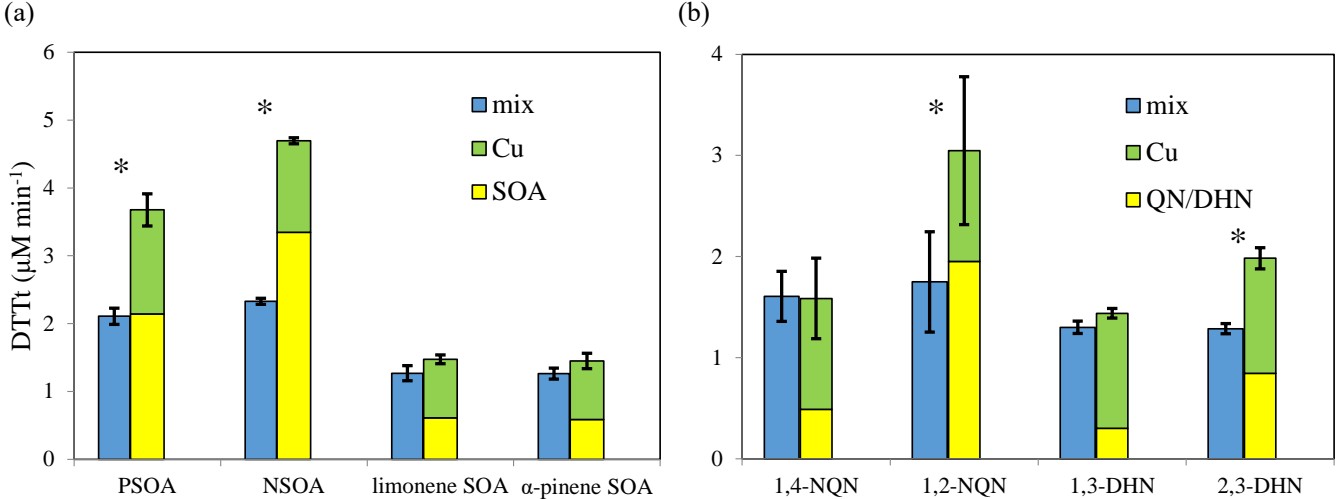

**Figure 8.** Significant OP depletions were observed when PSOA(43±4%), NSOA(50±6%), 1,2-NQN(42±7%) and 2,3-DHN(35±1%) mixed with Cu (II). The asterisk indicates significant difference between a pair of bars at a 95% confidence level.



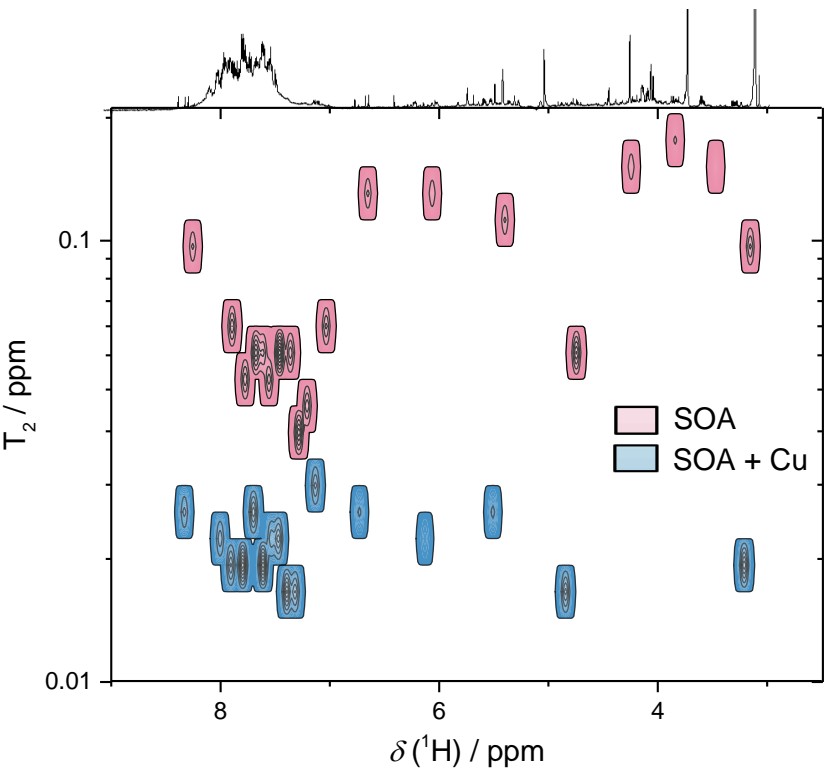

**Figure 9.** 2D ${}^1$H-NMR T2 relaxation contour map for NSOA with (blue) and without copper (red) with 1D NMR projections from the top. A general decreasing trend in T2 is observed here, which indicates interactions (binding) of Cu (II) with many NSOA components.



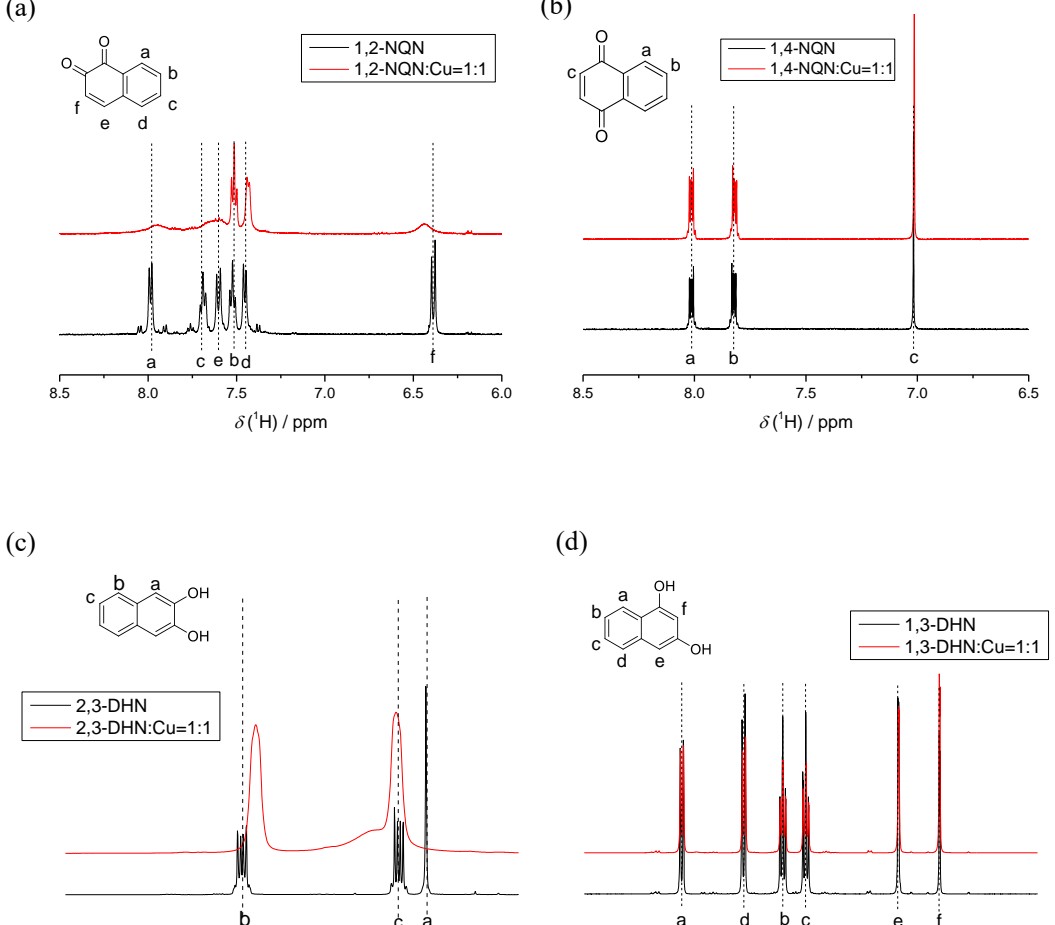

**Figure 10.** 1 D ¹H-NMR spectra of  (a) 1,2-NQN  (b) 1,4-NQN (c) 2,3-DHN (d)1,3-DHN and

their mixture with 1:1 ratio of Cu (II). Both 1,2-NQN and 2,3-DHN show the broadening of ¹H-

NMR peaks (protons at a, c, e, f and a, b, c for 1,2-NQN and 2,3-DHN, respectively) after

mixing with Cu(II).





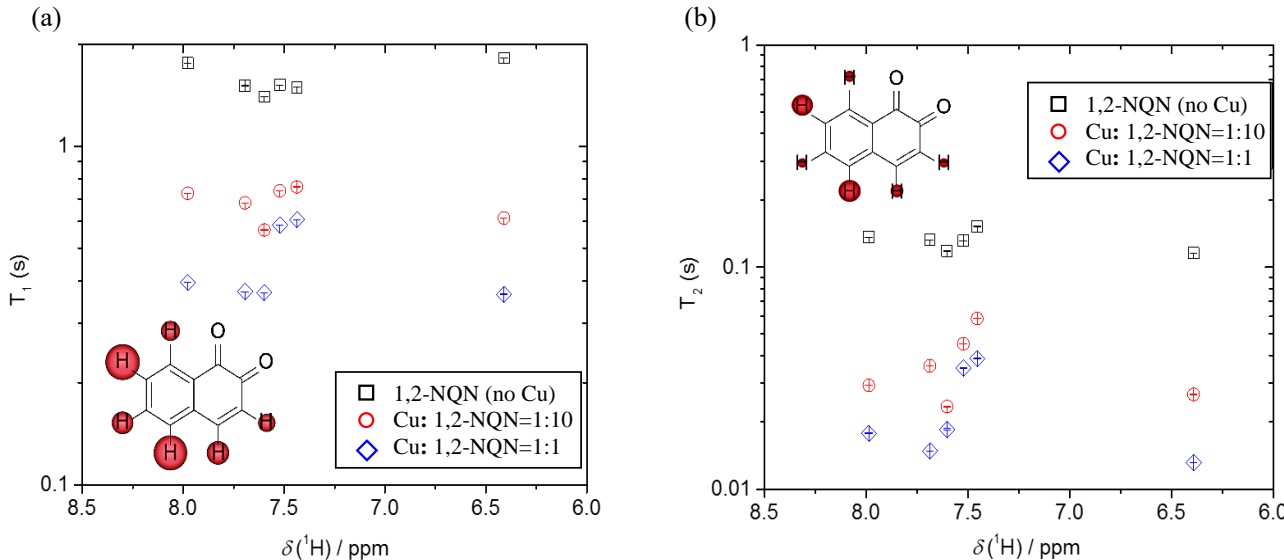

**Figure 11.** [1]H NMR relaxometry analyses for (a) T1, (b) T2 of 1,2-NQN mixed with Cu (II) at

different ratios: 1,2-NQN with no Cu (**black**),Cu(II):1,2-NQN = 1:10 (**red**), Cu(II):1,2-NQN

=1:1(**blue**). Both T1, T2 decreased when Cu was introduced into the system, indicating a smaller

scale of nuclear spin dynamic resulted from organic-metal binding. The molecular epitopes

illustrate the influence of Cu (II) binding on individual proton. A smaller sphere shadow on a

proton denotes a larger relaxation influence from Cu (II)-organic binding.