# Peer review of "Relationship between chemical composition and oxidative potential of secondary organic aerosol from polycyclic aromatic hydrocarbons"

_Atmospheric Chemistry and Physics, 2017_

## Referee Comment (RC1) · Anonymous Referee #1 · 4 Dec 2017

**Comments on "Relationship between Chemical Composition and Oxidative Potential of Secondary Organic Aerosol from Polycyclic Aromatic Hydrocarbons"**

In this manuscript, the authors assessed the DTT activities of lab-generated SOA using flow tubes from two PAHs (phenanthrene and naphthalene). The mass normalized DTT activities of SOA were compared to those from monoterpenes (limonene and α-pinene). Peroxide content was also determined to assess the contribution from peroxide to OP. Aging effects from oligomerization, heterogeneous oxidation, and mixing with copper were investigated on the DTT activities of SOA. In my opinion, this is an important work. It provides reference to future studies on the DTT activities of SOA and addresses some current open questions regarding OP. It provides a new angle of looking at chemical composition that contributes to OP and health. The manuscript overall is fairly strong and I recommend acceptance of the manuscript after following small corrections are done.

1. Line 36-37, the authors stated "Oxidative stress has been…, and is often expressed as the oxidative potential (OP)". This sentence is misleading in that "oxidative stress" is "oxidative potential" while these two terms are different. The authors should make sure the definitions of these two terms are clear.

2. Line 37-38, delete "mass normalized". OP can be expressed in "volume normalized" as well.

3. Section 2.3, the authors quoted McWhinney et al. papers for the DTT protocols, despite that, detailed protocols should be provided either in method section or the supplement. For example, what is the volume of sample added to each well, what is the concentration of DTT solution (i.e. the initial DTT concentration in the reaction), concentration of DTNB, was DTT consumed more than 50%, and etc.

4. Line 173, the SOA was extracted in Methanol then blown down to complete dryness before re-dissolving in phosphate buffer. Have the authors assessed the vaporization loss of SOA upon complete dryness? Some Methanol-soluble compounds might not be solubilized in phosphate buffer which is mainly DI. The authors could in a way underestimate the OP of SOA due to artifacts from complete dryness.

5. Line 188, should be "Fig. S2(b)"?

6. Line 363, peroxide content from α-pinene are 40-100% in this study which is larger than other previous studies (Docherty et al., 2005; Epstein et al., 2014; Mertes et al., 2012) where roughly 20-60% of peroxide are found in α-pinene SOA. Please explain why such large variation in this work and larger values compared to other studies.

7. Line 379, please explain why the number of DTTm of benzoyl peroxide is 38 pmol/min/ug while that in Table 2 is 160 pmol/min/ug.

8. The initial DTT at t=0 (18000 pmol) in various types of peroxides in Figure 3 are different from that in blank controls (~37000 pmol, Figure S2). Please explain why the authors chose different initial DTT amount to begin with between blanks and samples.

9. Figure 6, asterisk missing?

10. Figure 9, can the authors comment on the delta decrease of T2 relaxation larger in the lower $\delta(^1H)$ range than higher range with Cu added to SOA compared to SOA alone? This is not observed in 1,2-NQN case (Figure 11).

Docherty, K. S.; Wu, W.; Lim, Y. B.; Ziemann, P. J. Contributions of organic peroxides to secondary aerosol formed from reactions of monoterpenes with O3. Environ. Sci. Technol. 2005, 39(11), 4049−4059.

Epstein, S. A.; Blair, S. L.; Nizkorodov, S. A. Direct photolysis of α-pinene ozonolysis secondary organic aerosol: effect on particle mass and peroxide content. Environ. Sci. Technol. 2014, 48 (19), 11251−11258.

Mertes, P.; Pfaffenberger, L.; Dommen, J.; Kalberer, M.; Baltensperger, U. Development of a sensitive long path absorption photometer to quantify peroxides in aerosol particles (PeroxideLOPAP). Atmos. Meas. Tech. 2012, 5 (10), 2339−2348.

---

## Referee Comment (RC2) · Anonymous Referee #2 · 6 Dec 2017

Review for "Relationship between Chemical Composition and Oxidative Potential of Secondary Organic Aerosol from Polycyclic Aromatic Hydrocarbons" by Shunyao Wang et al.

**General comments**:

The authors of current manuscript investigated the correlation of chemical composition of laboratory generated naphthalene and phenanthrene SOA (NSOA and PSOA) with their oxidative potentials (OP) using dithiothreitol (DTT) assay in combination with LC-MS and NMR techniques. They found the oligomer-rich fractions but not the peroxides dominate the OP activity of NSOA and PSOA. Furthermore, they found the ozonolysis of NSOA particles can elevate their OP prominently. Later on, they found the DTT activities of the mixtures of copper ions with redox-active organics or SOA are not additive. Based on NMR measurement, the authors assigned this phenomenon to the formation of complexes. Overall the presented results are interesting and the scientific is sound. The manuscript was written well. Therefore I would like to recommend this manuscript to be published in *Atmos. Chem. Phys.* if my following concerns can be fully addressed.

**Specific comments**:

1. In Figure 1, the authors illustrated that both of hydroquinone and semiquinone can reduce $O_2$ to $O_2^-$. However, Dellinger et al. (*Chem. Res. Toxicol.*, 14, 1371-1377, 2001.) suggested that semiquinone is responsible for reducing $O_2$ to $O_2^-$, but hydroquinone is responsible for transforming $O_2^-$ to $H_2O_2$. Is there any conflict of Figure 1 with literature?

2. In lines of 161 to 163, the authors said ''Within 3 days of collection, the filters were extracted in methanol (HPLC grade, 99.9%, Sigma Aldrich, St. Louis, MO, USA), by ultra-sonication at room temperature for more than 3 minutes.''. Respect to this experiment procedure, I have two questions. Firstly, Krapf et al. (*Chem*, 1, 603-616, 2016.) demonstrated that 'OOH-containing molecules are labile and decay with a half-life of only 45±3 min'. So the aging of SOA in freezer for 3 days may significantly decrease the final OP of them? Secondly, the authors extracted SOA into methanol and then measured their OP with DTT assay. Considering organic solvent has different effect from water to

influence the OP of ambient particulate matters (Yang et al., Atmos. Environ. 2014), I am wondering how significant the methanol and ultra-sonication operation will influence the OP of SOA here.

3. In line 171: the author said ''....DTT, an antioxidant that...''. This is a wrong description. DTT is normally used as a surrogate of biological reductant (NADPH etc.), but itself is not antioxidant (Charrier and Anastasio. *Atmos. Chem. Phys.*, 12, 9321-9333, 2012. Shiraiwa et al., *Environ. Sci. Technol.*, 2017, 51 (23), pp 13545-13567.).

4. In line 173-174, the authors said 'The SOA extracts were first evaporated to complete dryness in a 5.0 L min$^{-1}$ of N$_2$ using a blow off system (N-EVAP, Organomation, USA).' Then in lines of '468-470', the authors indicated that 'The overall increased volatility may lead to evaporation of smaller redox-active molecules and decrease the DTT$_t$ compared to the N$_2$ exposure group.' So whether the loss of small molecular redox active compounds have also happened during the evaporation of SOA, and how significant this process will influence the DTT of SOA especially the monomer rich fraction?

5. In Fig 5 c and d, the sub-captain is 'contribution to NSOA DTT activity', but the pie charts actually showed the relative total DTT decay rate (DTT$_t$) of different N/PSOA fractions. The misleading word 'contribution' here is different from the one used in line 336 of the manuscript, which is based on 1, 2- and 1, 4-naphthoquinone particulate concentrations exactly (McWhinney et al. 2013, which is DTT$_m$). Considering the current study cannot quantify the recovery of monomer-rich and oligomer-rich compounds from N/PSOA (as stated by the authors in lines 410 to 412), the authors should not be able to use mass normalized DTT value to predict the contribution of monomer-rich and oligomer-rich fractions to DTT$_m$ of N/PSOA. The caption and relevant illustration for Figure 5 c and d should be improved and clarified clearly, especially to compare with the work by McWhinney et al.

6. Respect to the results discussed in section 3.5 especially that showed in the Figure 8, whether the Cu initiated Fenton like reactions or relevant redox chemistry also play significant role? The authors are encouraged to discuss this aspect in the manuscript.

7. It will be useful to add the averaged carbon oxidation state values of monomer and oligomer rich fractions of N/PSOA into Figure 6.

8. The section of 'References' should be improved carefully, e.g. the references of Mentel et al., 2015 (in line 347) and Tong et al., 2016 (in line 383) could not be find in reference list. In addition, in line 740: '2009' should be '2010'. In line 680, the reference of 'Di Lorenzo et al., *Geophys. Res. Lett.*, 43, 458-465, 2016' should be separated with the previous one.

9. Typos should be corrected for the whole manuscript, e.g. blank space should be used between number and unit: '160W' should be '160 W'.

---

## Author Comment (AC1) · 10 Feb 2018

**Response to comments from referee #1**

We would like to thank the reviewer for the helpful comments and suggestions.

Our response and corresponding modifications are listed below.

**General comments:**

In this manuscript, the authors assessed the DTT activities of lab-generated SOA using flow tubes from two PAHs (phenanthrene and naphthalene). The mass normalized DTT activities of SOA were compared to those from monoterpenes (limonene and α-pinene). Peroxide content was also determined to assess the contribution from peroxide to OP. Aging effects from oligomerization, heterogeneous oxidation, and mixing with copper were investigated on the DTT activities of SOA. In my opinion, this is an important work. It provides reference to future studies on the DTT activities of SOA and addresses some current open questions regarding OP. It provides a new angle of looking at chemical composition that contributes to OP and health. The manuscript overall is fairly strong and I recommend acceptance of the manuscript after following small corrections are done.

**Specific comments:**

**1.** Line 36-37, the authors stated "Oxidative stress has been…, and is often expressed as the oxidative potential (OP)". This sentence is misleading in that "oxidative stress" is "oxidative potential" while these two terms are different. The authors should make sure the definitions of these two terms are clear.

**Response:** Oxidative potential is the capacity for inhaled air pollutants to cause redox imbalance through consumption of antioxidants and generation of reactive oxygen species (ROS).

Oxidative stress is the redox imbalance induced by oxidative potential (Adler et al., 1999;Finkel and Holbrook, 2000). Hence, the definitions of oxidative potential and oxidative stress in this study were modified as below:

"Oxidative stress has been proposed as one of the main mechanisms for PM toxicity in recent years, and is caused by oxidative potential (OP) (Li et al., 2003b). OP is exhibited as the capacity of inhaled PM to induce oxidative stress, the redox imbalance generated through consumption of antioxidants and production of reactive oxygen species (ROS) (Anti ñolo et al., 2015; Shen et al., 2011; Shiraiwa et al., 2012)."

**2.** Line 37-38, delete "mass normalized". OP can also be expressed in "volume normalized".

**Response:** Modified. Please see line 38 in the manuscript.

**3.** Section 2.3, the authors quoted McWhinney et al. papers for the DTT protocols, despite that, detailed protocols should be provided either in method section or the supplement. For example, what is the volume of sample added to each well, what is the concentration of DTT solution (i.e. the initial DTT concentration in the reaction), concentration of DTNB, was DTT consumed more than 50%, and etc.

**Response:** The method section has been expanded to include in more experimental details. Please see line 179-line 189 and Fig. S2c, d in the manuscript.

The volume of SOA sample in each well was 160 μL. The concentration of DTT was 0.2 mM. The concentration of DTNB was 2 mM (10 times in excess) of DTT. Such initial DTT concentration was determined in order to maintain an eventual DTT consumption near/over 50% (Fig. S2c, d) for PAH derived SOA tested in our study.

[Figure]

**Figure S2.** (c) DTT activity of PSOA. (d) DTT activity of NSOA.

**4.** Line 173, the SOA was extracted in Methanol then blown down to complete dryness before re-dissolving in phosphate buffer. Have the authors assessed the vaporization loss of SOA upon complete dryness? Some Methanol-soluble compounds might not be solubilized in phosphate buffer which is mainly DI. The authors could in a way underestimate the OP of SOA due to artifacts from complete dryness.

**Response:** This is an important question for oxidative potential assays. In this work, we determined "complete dryness" based on visual inspection. When using the $N_2$ blow-off system to evaporate methanol, other compounds with relative high volatilities may also evaporate during this process. Based on the reviewer's suggestion, we conducted a series of tests to investigate whether the loss of volatile compounds along with the $N_2$ blow-off procedure affects the DTT activity of SOA. Four sets of NSOA methanol solutions (with the same SOA amount extracted) were evaporated under the $N_2$ blow-off system to various extents, with which 100%, 66%, 33%, <1% of the original methanol remained. All the solutions were then replenished with methanol to the same total volume and reconstituted to the experimental concentration with phosphate buffer for measurement of DTT activity. The results are shown below:

[Figure]

Despite the difference in degree of evaporation, there was no significant change in the $DTT_m$ observed in this study. Thus, we deduced the evaporation of volatile SOA compounds during $N_2$ blow-off procedure did not lead to a significant underestimation of the OP of SOA in this study. It is likely that SOA compounds from naphthalene/ phenanthrene with high volatility may be less

oxygenated and less likely to be dominant OP contributor, which is also consistent with our findings in section 3.4. However, it should be noted that this conclusion may not hold for all atmospheric samples, and the contribution of semivolatile compounds to oxidative potential may be significant and could be underestimated with the extraction protocols described in this work.

**5.** Line 188, should be "Fig. S2(b)"?

**Response:** Corrected. Please find line 191 in the manuscript.

**6.** Line 363, peroxide content from α-pinene are 40-100% in this study which is larger than other previous studies (Docherty et al., 2005; Epstein et al., 2014; Mertes et al., 2012) where roughly 20-60% of peroxide are found in α-pinene SOA. Please explain why such large variation in this work and larger values compared to other studies.

**Response:** Compared to many of the previous studies which used relatively small molecules (i.e. $H_2O_2$ or -O-O-) for estimating SOA peroxide content (Nguyen et al., 2010;Epstein et al., 2014), the mass of SOA peroxides was calibrated by using benzoyl peroxide as standard in our study. The molecular weight of benzoyl peroxide (242.23 g mol$^{-1}$) might be larger than the actual averaged molecular mass of peroxides in SOA, leading to a relative higher result. In addition, Mertes et al. (2012) assumed spherical particle geometry density of α-pinene SOA 1.3 g cm$^{-3}$ while our study used a density of 1.25 g cm$^{-3}$. Using a smaller SOA density might also lead to a higher SOA peroxide fraction. Moreover, according to the study of Mutzel et al. (2013), the process of ultra-sonication might also lead to the decomposition of organic peroxides into

hydroxy radical. Those formed hydroxyl radicals are able to further form $H_2O_2$ and inflate the final SOA peroxide content. At the same time, other studies have shown that organic peroxides decompose into carbonyls and alcohols, which might also result in a lower detected peroxide content. Compare to the study of Docherty et al. (2005), the ultra-sonication time of our study was under 5 minutes while they used 10 minutes, which may explain the differences in results if ultrasonication leads to overall loss of peroxide content. Other factors like the $O_3$ exposure, temperature and relative humidity during SOA formation may also lead to differences in the calculated peroxide content. Overall, the conclusion from our study is that despite the relatively high peroxide content of α-pinene SOA in our study and in other studies, there is no apparent contribution to DTT activity, suggesting that organic peroxides are not dominant SOA OP contributors in the systems we studied.

**7.** Line 379, please explain why the number of $DTT_m$ of benzoyl peroxide is 38 pmol/min/ug while that in Table 2 is 160 pmol/min/ug.

**Response:** Thanks for pointing this out. The value of "160" was a typo. After various testing, the $DTT_m$ of benzoyl peroxide in our study was eventually measured to be 37 pmol $min^{-1}$ $μg^{-1}$. Corresponding information was corrected in the manuscript. Please find the updated Fig.3, Table 2 and line 382 in the manuscript.

**8.** The initial DTT at t=0 (18000 pmol) in various types of peroxides in Figure 3 are different from that in blank controls (~37000 pmol, Figure S2). Please explain why the authors chose different initial DTT amount to begin with between blanks and samples.

**Response:** The $DTT_m$ for peroxides are generally lower than other tested chemicals (redox-active) in this study. We determined this initial DTT concentration (in excess) for peroxide based on an estimation of eventual DTT consumption percentage. However, in order to measure DTT activity under the same initial conditions, we repeated the OP measurement of peroxides (120 µM) with the same initial DTT concentration (0.2 mM) as other DTT experiments conducted in this study, such that they are comparable. The results are shown in the updated Fig.3.

[Figure]

**Figure 3.** DTT activity of various types of peroxides (hydrogen peroxide, cumene peroxide, tert-Butyl peroxide, benzoyl peroxide). With the same initial concentration of peroxide (120 µM), benzoyl peroxide has the highest DTT activity (converted to $DTT_m$ of 37 pmol min$^{-1}$ ug$^{-1}$).

**9.** Figure 6, asterisk missing?

**Response:** Corrected. Asterisks were added onto Fig.6.

**10.** Figure 9, can the authors comment on the delta decrease of T2 relaxation larger in the lower δ($^1$H) range than higher range with Cu added to SOA compared to SOA alone?

This is not observed in 1,2-NQN case (Figure 11)

**Response:** Thank you for the comment. We apologize for not clarifying the regions in $^1$H NMR of the SOA data. The peaks at lower δ ($^1$H) region (3-5ppm) are mainly associated with protons on aliphatic hydroxyl, while those from 6-9 ppm are mainly from aromatics. As quinones (1,2-NQN) in this study is purely aromatic, there are no groups that resonate lower than 6 ppm. We have now updated it in the manuscript to clarify the NMR regions.

We clarified the description in the revised manuscript (section 3.5) as below:

"……interactions between copper and SOA components. In general, protons adjacent to and within aliphatic hydroxyl groups resonate between 3-5 ppm, while aromatic groups and double bonds arise between 5-9 ppm. Before the addition of copper, NSOA T2 relaxation time show a range of values consistent with a complex material with a diversity of functional groups and dynamics. It was after the addition of copper, both of the aliphatic hydroxyl and aromatic region showed a decrease in T2 and inherit roughly similar relaxation values. This suggested copper is able to influence a wide range of SOA components, with both structural categories (aliphatic hydroxyl and aromatic) involved in copper binding to some extent."

**References**

Adler, V., Yin, Z., Tew, K. D., and Ronai, Z. e.: Role of redox potential and reactive oxygen species in stress signaling, Oncogene, 18, 1999.

Docherty, K. S., Wu, W., Lim, Y. B., and Ziemann, P. J.: Contributions of organic peroxides to secondary aerosol formed from reactions of monoterpenes with $O_3$, Environ. Sci. Technol., 39, 4049-4059, 2005.

Epstein, S. A., Blair, S. L., and Nizkorodov, S. A.: Direct photolysis of α-pinene ozonolysis secondary organic aerosol: effect on particle mass and peroxide content, Environ. Sci. Technol., 48, 11251-11258, 2014.

Finkel, T., and Holbrook, N. J.: Oxidants, oxidative stress and the biology of ageing, Nature, 408, 239-247, 2000.

Mertes, P., Pfaffenberger, L., Dommen, J., Kalberer, M., and Baltensperger, U.: Development of a sensitive long path absorption photometer to quantify peroxides in aerosol particles (Peroxide-LOPAP), Atmos. Meas. Tech., 5, 2339, 2012.

Mutzel, A., Rodigast, M., Iinuma, Y., B öge, O., and Herrmann, H.: An improved method for the quantification of SOA bound peroxides, Atmos. Environ., 67, 365-369, 2013.

Nguyen, T. B., Bateman, A. P., Bones, D. L., Nizkorodov, S. A., Laskin, J., and Laskin, A.: High-resolution mass spectrometry analysis of secondary organic aerosol generated by ozonolysis of isoprene, Atmos. Environ., 44, 1032-1042, 2010.

---

## Author Comment (AC2) · 10 Feb 2018

**Response to comments from referee #2**

We appreciate the valuable and thoughtful comments provided by the referee.

Our response and corresponding modifications are listed below.

**General comments:**

The authors of current manuscript investigated the correlation of chemical composition of laboratory generated naphthalene and phenanthrene SOA (NSOA and PSOA) with their oxidative potentials (OP) using dithiothreitol (DTT) assay in combination with LC-MS and

NMR techniques. They found the oligomer-rich fractions but not the peroxides dominate the OP

activity of NSOA and PSOA. Furthermore, they found the ozonolysis of NSOA particles can elevate their OP prominently. Later on, they found the DTT activities of the mixtures of copper ions with redox-active organics or SOA are not additive. Based on NMR measurement, the authors assigned this phenomenon to the formation of complexes. Overall the presented results are interesting and the scientific is sound. The manuscript was written well. Therefore

I would like to recommend this manuscript to be published in Atmos. Chem. Phys. if my following concerns can be fully addressed.

**Specific comments:**

**1.** In Figure 1, the authors illustrated that both of hydroquinone and semiquinone can reduce

$O_2$ to $O_2 \cdot^-$. However, Dellinger et al. (Chem. Res. Toxicol., 14, 1371-1377, 2001.) suggested that semiquinone is responsible for reducing $O_2$ to $O_2^{\cdot -}$, but hydroquinone is responsible for transforming $O_2^{\cdot -}$ to $H_2O_2$. Is there any conflict of Figure 1 with literature?

**Response:** Thank you very much for this comment. Redox-active quinones act as electron transfer agent that can constantly transfer electrons from reductants to oxidants (e.g. from

NADPH to $O_2$). The redox-chemistry for quinones inside human body is relative complicated through both enzymatic and nonenzymatic redox cyclings accompanied by the generation of

ROS while Fig.1 presented in our manuscript is highly simplified. Base on the previous studies, quinones at a reduced state (semiquinone and hydroquinone) are able to be oxidized by monooxygenase or peroxidase enzymes, molecular oxygen (autoxidation) and metal ions (Roginsky et al., 1999;Monks et al., 1992;Bolton et al., 2000). Transfering electrons from $O_2^{\cdot -}$ to

$H_2O_2$ has been proved to be an alternative pathway for hydroquinone to evolve into semiquinone/quinone.

**2.** In lines of 161 to 163, the authors said "Within 3 days of collection, the filters were extracted in methanol (HPLC grade, 99.9%, Sigma Aldrich, St. Louis, MO, USA), by ultrasonication at room temperature for more than 3 minutes." Respect to this experiment procedure, I have two questions. Firstly, Krapf et al. (Chem, 1, 603-616, 2016.)

demonstrated that 'OOH-containing molecules are labile and decay with a half-life of only

45±3 min'. So the aging of SOA in freezer for 3 days may significantly decrease the final

OP of them? Secondly, the authors extracted SOA into methanol and then measured their

OP with DTT assay. Considering organic solvent has different effect from water to influence the OP of ambient particulate matters (Yang et al., Atmos. Environ. 2014), I am wondering how significant the methanol and ultra-sonication operation will influence the

OP of SOA here.

**Response:** Krapf et al. (2016) showed OOH-containing molecules are labile and decay with a half-life of only 45 ±3 min. This SOA peroxide thermal-decomposition fate was investigated under room temperature. In this work, the SOA samples were stored under -20 ℃ after being collected and wrapped in prebaked aluminum foil. In the study of Jiang et al. (2017), they have tested the DTT response against the stability of peroxide. They stored organic peroxide and hydrogen peroxide under room temperature and 4 °C. The DTT responses were maintained above 90% within 3 days under both temperatures. Moreover, peroxide stored under 4 °C

exhibited a higher DTT response than that of room temperature, indicating lower storage temperature could prevent the thermal decomposition of organic peroxide. Admittedly, the peroxide content in SOA is labile, but the stability of SOA peroxide under the storage condition of our study (-20 ℃) seems unlikely to make a significant difference in the DTT results.

Based on previous studies (McWhinney et al., 2011), quinones are the major OP contributors in

PAH derived SOA and their solubility in methanol are generally higher than in MQ water. Also

Yang et al. (2014) suggested that quartz filter absorbs water during extraction which might bring about the loss of SOA. As a result, we consider using methanol as our SOA extraction solvent and then reconstitute with phosphate buffer for DTT analysis. The effects on SOA composition coming from ultra-sonication has been discussed by Mutzel et al. (2013), which found ultra- sonication might elevate the peroxide content (thermal-lability) inside SOA. Despite the higher measured peroxide content, there is no associated change in the OP of SOA. We therefore conclude that we find no evidence of association between peroxide content and OP in NSOA.

**3.** In line 171: the author said "….DTT, an antioxidant that…". This is a wrong description. DTT

is normally used as a surrogate of biological reductant (NADPH etc.), but itself is not antioxidant (Charrier and Anastasio. Atmos. Chem. Phys., 12, 9321-9333, 2012. Shiraiwa et al., Environ. Sci.

Technol., 2017, 51 (23), pp 13545-13567.).

**Response:** This sentence has been revised as "……DTT, an antioxidant surrogate that……" in line 171.

**4.** In line 173-174, the authors said "The SOA extracts were first evaporated to complete dryness in a 5.0 L min$^{-1}$ of $N_2$ using a blow off system (N-EVAP, Organomation, USA)." Then in lines of 468-470, the authors indicated that "The overall increased volatility may lead to evaporation of smaller redox-active molecules and decrease the $DTT_t$ compared to the

$N_2$ exposure group." So whether the loss of small molecular redox active compounds have also happened during the evaporation of SOA, and how significant this process will influence the DTT of SOA especially the monomer rich fraction?

**Response:** Thank you for pointing this out. The $N_2$ blow-off procedure is for SOA methanol extracts while the later conclusion of "carbon loss due to volatility" was based on the EC/OC

results after heterogeneous ozonolysis of SOA filter.  The duration for heterogeneous oxidation ranges from 1 hour to 24 hours in our study while the $N_2$ blow off procedure for SOA methanol extracts only lasts for 30-40 minutes. Also the evaporation rate may be lower in a dilute methanol solution. However, there may still be potential volatile losses of low molecular weight high volatility SOA compounds during the $N_2$ blow off procedure. To assess the potential impacts on the DTT activity, we evaporated SOA methanol extracts under the N2 blow-off

system to various extents, with which 100%, 66%, 33%, <1% of methanol were remained. All the solutions were then reconstituted the same experimental concentration by adding methanol and phosphate buffer. There was no significant difference among each experimental group. Thus, we conclude the evaporation of high volatile SOA compounds during N2 blow-off procedure is not likely to cause a significant underestimation of the OP studied here.

**5.** In Fig 5 c and d, the sub-captain is "contribution to NSOA DTT activity", but the pie charts actually showed the relative total DTT decay rate ($DTT_t$) of different N/PSOA fractions. The misleading word "contribution" here is different from the one used in line 336 of the manuscript, which is based on 1, 2- and 1, 4-naphthoquinone particulate concentrations exactly (McWhinney et al. 2013, which is $DTT_m$). Considering the current study cannot quantify the recovery of monomer-rich and oligomer-rich compounds from N/PSOA (as stated by the authors in lines 410 to 412), the authors should not be able to use mass normalized DTT value to predict the contribution of monomer-rich and oligomer-rich fractions to $DTT_m$ of N/PSOA. The caption and relevant illustration for Figure 5 c and d should be improved and clarified clearly, especially to compare with the work by

McWhinney et al.

**Response:** We apologize for not clarifying this.

Line 10 in Abstract has been modified as "…dominates DTT activity in both SOA systems…"

Text in section 3.3 has been revised to "… in Fig. 5, both the monomer-rich and oligomer-rich fractions are reactive towards DTT. Compared to the $DTT_t$ of NSOA before LC separation, the relative $DTT_t$ of monomer-rich fraction and oligomer-rich fraction were 16±3% and 56±10%, respectively (Fig. 5c). For PSOA, relative $DTT_t$ from monomer-rich fraction and oligomer-rich fraction were 40±8% and 50±5%, respectively (Fig. 5d)."

The sub-caption for Fig. 5c and d has been revised to "Relative $DTT_t$ from monomer-rich fraction and oligomer-rich fraction in NSOA (c) and PSOA (d) systems."

**6.** Respect to the results discussed in section 3.5 especially that showed in the Figure 8, whether the Cu initiated Fenton like reactions or relevant redox chemistry also play significant role? The authors are encouraged to discuss this aspect in the manuscript

**Response:** Thanks for this suggestion. First, we observed that the OP reduction of quinones in our study has a structural dependence. It was with 1,2-NQN and 2,3-DHN that we witnessed a significant OP depletion but not with 1,4-NQN and 1,3-DHN. Similar OP depletion trend should be observed with the four quinones after Cu was added to the system if Cu initiated Fenton like reactions or other relevant redox chemistry also played significant roles in the OP depletion.

While the Fenton reaction is faster under acidic conditions (Zepp et al., 1992;Kang and Hwang,

2000), we would expect the additional OH ·or HOO ·produced through the Fenton reaction under controlled pH conditions to elevate the OP level while this is inconsistent with what we observed with 1,2-NQN, 2,3-DHN and PAH derived SOA. Combining the NMR organic molecular structural information, we propose that the OP depletion in our study is more likely due to the binding between organics and Cu.

**7.** It will be useful to add the averaged carbon oxidation state values of monomer and oligomer rich fractions of N/PSOA into Figure 6.

**Response:** Thanks for this advice. Since most of the m/z peaks in ESI-MS spectra (negative mode) are polar compounds due to the electron spray ionization. The ESI-MS signal weighed

$\overline{OS}c$ of both monomer and oligomer would not be comprehensive and representative enough for the total SOA system we studied here. Also, the calculation of $\overline{OS}c$ relies on the molecular formula of each compound while the accuracy of peak fitting for higher m/z range (oligomer)

might be lower than that of the lower m/z range (monomer).  So far, we are currently unable to add accurate $\overline{OS}$c values for monomer and oligomer rich fractions of N/PSOA.

**8.** The section of 'References' should be improved carefully, e.g. the references of Mentel et al., 2015 (in line 347) and Tong et al., 2016 (in line 383) could not be find in reference list.

In addition, in line 740: '2009' should be '2010'. In line 680, the reference of 'Di Lorenzo et al., Geophys. Res. Lett., 43, 458-465, 2016' should be separated with the previous one.

**Response:** References of this manuscript has been checked and corrected.

**9.** Typos should be corrected for the whole manuscript, e.g. blank space should be used between number and unit: '160W' should be '160 W'

**Response:** Typos of this manuscript has been checked and corrected.

**References**

Bolton, J. L., Trush, M. A., Penning, T. M., Dryhurst, G., and Monks, T. J.: Role of quinones in toxicology, Chem. Res. Toxicol., 13, 135-160, 2000.

Jiang, H., Jang, M., and Yu, Z.: Dithiothreitol activity by particulate oxidizers of SOA produced from photooxidation of hydrocarbons under varied NOx levels, Atmos. Chem. Phys., 17, 9965-

9977, 2017.

Kang, Y. W., and Hwang, K. Y.: Effects of reaction conditions on the oxidation efficiency in the

Fenton process, Water Res., 34, 2786-2790, 2000.

Krapf, M., El Haddad, I., Bruns, Emily A., Molteni, U., Daellenbach, Kaspar R., Prévôt, André S.

H., Baltensperger, U., and Dommen, J.: Labile Peroxides in Secondary Organic Aerosol, Chem,

1, 603-616, http://dx.doi.org/10.1016/j.chempr.2016.09.007, 2016.

McWhinney, R. D., Gao, S. S., Zhou, S., and Abbatt, J. P. D.: Evaluation of the effects of ozone oxidation on redox-cycling activity of two-stroke engine exhaust particles, Environ. Sci.

Technol., 45, 2131-2136, 2011.

Monks, T. J., Hanzlik, R. P., Cohen, G. M., Ross, D., and Graham, D. G.: Quinone chemistry and toxicity, Toxicol. Appl. Pharm., 112, 2-16, 1992.

Mutzel, A., Rodigast, M., Iinuma, Y., Böge, O., and Herrmann, H.: An improved method for the quantification of SOA bound peroxides, Atmos. Environ., 67, 365-369, 2013.

Roginsky, V. A., Pisarenko, L. M., Bors, W., and Michel, C.: The kinetics and thermodynamics of quinone–semiquinone–hydroquinone systems under physiological conditions, J. Chem. Soc.,

Perkin Transactions 2, 871-876, 1999.

Yang, A., Jedynska, A., Hellack, B., Kooter, I., Hoek, G., Brunekreef, B., Kuhlbusch, T. A.,

Cassee, F. R., and Janssen, N. A.: Measurement of the oxidative potential of PM $_{2.5}$ and its constituents: the effect of extraction solvent and filter type, Atmos. Environ., 83, 35-42, 2014.

Zepp, R. G., Faust, B. C., and Hoigne, J.: Hydroxyl radical formation in aqueous reactions (pH

3-8) of iron (II) with hydrogen peroxide: the photo-Fenton reaction, Environ. Sci. Technol., 26,

313-319, 1992.